# Improved Accuracy and Efficiency of Flood Inundation Mapping of Low-, Medium-, and High-Flow Events Using the AutoRoute Model

Michael L. Follum[1,2], Ricardo Vera[3], Ahmad A. Tavakoly[1,4], and Joseph L. Gutenson[1,5]

[1]Coastal and Hydraulics Laboratory, Engineer Research and Development Center, 3909 Halls Ferry Road, Vicksburg, MS 39180, USA
[2]Wyoming Area Office, U.S. Bureau of Reclamation, 705 Pendell Blvd., Mills, WY 82644, USA.
[3]Cold Regions Research and Engineering Laboratory, Engineer Research and Development Center, 72 Lyme Road, Hanover, NH 03755, USA
[4]Earth System Science Interdisciplinary Center, University of Maryland, College Park, MD 20740, USA
[5]National Water Center, National Oceanic and Atmospheric Administration, 205 Hackberry Ln, Tuscaloosa, AL 35401, USA

*Correspondence to*: Michael L. Follum (follumm@gmail.com)

**Abstract.** This article presents improvements and development of a post-processing module for the regional scale flood mapping tool, AutoRoute. The accuracy of this model to simulate low, medium, and high flow rate scenarios is demonstrated at seven test sites within the U.S. AutoRoute is one of the tools used to create high-resolution flood inundation maps at regional- to continental-scales, but has previously only been tested using extreme flood events. Modifications to the AutoRoute model and post-processing scripts are shown to improve accuracy (e.g. average $F$ value increase of 17.5% for low-flow events) and computational efficiency (simulation time reduced by over 40%) when compared to previous versions. Although flood inundation results for low-flow events are shown to be comparable with published values (average $F$ value of 63.3%), the model results tend to be overestimated, especially in flatter terrain. Higher-flow scenarios tend to be more accurately simulated (average F value of 77.5%). With improved computational efficiency and the enhanced ability to simulate both low and high flow scenarios the AutoRoute model may be well suited to provide first-order estimates of flooding within an operational, regional- to continental-scale hydrologic modelling framework.

## 1 Introduction

Recent advances have demonstrated continental-scale flow forecasting models capable of simulating thousands of stream reaches simultaneously (e.g. National Water Model (NWM) (http://water.noaa.gov/about/nwm); Streamflow Prediction Tool (SPT) (Snow et al., 2016; Wahl 2016)). Although flow simulations at these scales are beneficial, water managers and emergency personnel benefit more from high-resolution flood inundation maps to make operational decisions (such as evacuation, road closures, etc.). Advanced hydraulic models typically operated from the reach-scale to the small-basin-scale have shown some success in simulating flood inundation at the continental scale (Wing et al., 2017), but at a high computational cost. Due to low data requirements, fast initial set-up times, and lower computational burden, lower-complexity hydraulic models have been developed in recent years to simulate flood inundation quickly using continental-scale hydrologic modelling

outputs. Although not meant to replace the higher-fidelity hydraulic models, these lower-complexity models can provide a reasonable first-order approximation of flood inundation over regional to continental extents and help prioritize where deployment of the higher-fidelity hydraulic models are needed (Follum et al., 2019). The National Oceanic and Atmospheric

Administration (NOAA) National Water Center (NWC) has adopted the Height Above Nearest Drainage (HAND) model (Liu et al., 2018; Zheng et al., 2016) to use in conjunction with the NWM within the U.S. Due to a need for connecting hydrologic data to mobility models for the military, the U.S. Army Coastal and Hydraulics Laboratory (CHL) developed the AutoRoute flood and mobility model (Follum, 2012; Follum et al., 2017; McKinley et al., 2012). AutoRoute works in conjunction with the SPT (Follum et al., 2017) to provide hydrologic and mobility guidance in data sparse environments outside the continental

United Stated (OCONUS). Currently SPT is run operationally (15-day streamflow forecasts updated twice daily) by CHL for approximately 70% of the world (between latitudes ~54°S and ~60°N based on data availability of the HydroSHEDS and HydroBASINS datasets (Lehner and Grill, 2013) from which SPT streamlines are derived). AutoRoute is currently operated in an ad-hoc basis when flood inundation or mobility assessments are required.

Both HAND and AutoRoute are raster-based models. Using the high-resolution National Hydrography Dataset Plus (NHDPlus) dataset (Horizon Systems Corporation, 2007; McKay et al., 2012) and a ~9 m digital elevation model (DEM), Liu et al. (2018) created HAND rasters for the entire U.S. A HAND raster simply shows the relative height of a cell above the nearest NHDPlus stream line (nearest in terms of drainage distance). Flow-depth rating curves are assigned to each stream reach (Zheng et al., 2018) so if given a flow rate the stage of the river can be calculated. Any HAND raster cell with a value

less than the calculated river stage is considered flooded (inundated). However, this process relies heavily on pre-computed flow-depth relationships currently not available for much of the world.

AutoRoute was initially developed by CHL to automatically develop cross-sections along ephemeral streams/rivers to assess gap-crossing capabilities of military vehicles during flood events (Follum, 2012; McKinley et al., 2012). Because AutoRoute

was utilized for ephemeral streams the channel profile (including bathymetry) was assumed to be represented by the DEM. Recently, AutoRoute has been applied with large-scale river routing models (such as the RAPID model (David et al., 2011; Tavakoly et al., 2017) within SPT) to simulate high-resolution (<30m spatial resolution) flood inundation maps over large extents: 230,000 km$^2$ area in the Midwest United States; 109,500 km$^2$ area in the Mississippi Delta (Follum et al., 2017); Sava River Basin; Puerto Rico (Follum et al., 2018); Navajo Nation (Follum et al., 2019); and Luzon, Philippines (Wahl et al.,

2017). Stream networks (polyline format) within the U.S. are defined using the NHDPlus dataset. Outside the U.S. stream networks (polyline format) for approximately 70% of the world have been created using HydroSHEDS and HydroBASINS datasets (see Snow et al., 2015 for an example). AutoRoute converts the polyline stream locations to a raster or table format (see Follum et al. (2017) for details). Cross-sections are automatically sampled for each stream cell from a DEM and the normal depth is then calculated for a given flow rate using Manning's equation. The extent and depth of flooding within the

cross-section is then mapped to a raster format. Only cells within the raster used for cross-sections will show flood extent or

depth. A post-processing step is often utilized where flood extent results in raster format are converted to a polygon format. The main purpose of the post-processing step is to overcome inaccuracies in the flood extents created by AutoRoute. Holes in the floodplain (cells not captured by cross-sections) are filled, the boundaries along the floodplain are smoothed, and outliers in the flood extent (cells that show flooding where no other surrounding cells show flooding) are omitted. Outliers in the flood

map are caused by large variations in flow depths along a given stream reach (Afshari et al., 2018; Follum et al. 2017), often caused by high elevation values due to bridges (Follum et al. 2017) or spikes in the DEM; cross-sections not being sampled perpendicular to the stream channel; and errors in calculating the slope of the channel (related to errors in the stream network or DEM). It is expected that these variations in depth and flood extent will be more pronounced in low-flow events where differences in depth or inundation extent may be more evident in an inundation map. Computationally, the post-processing

step takes almost as long as the execution of the AutoRoute model itself (Follum et al., 2017). Additionally, this post-processing step does not consider the terrain data; the post-processing is used only to make flood inundation maps appear more continuous.

Afshari et al. (2018) compared HAND, AutoRoute (with post-processing), and HEC-RAS 2D (USACE, 2016) at two locations:

Cedar River watershed in Iowa, and the Black Warrior River in Alabama. Three statistical flow conditions were tested at each site, the 10-, 100-, and 500-yr flow rates. The HAND and AutoRoute models produced similar flood inundation maps when compared to the more-advanced HEC-RAS 2D model, but both HAND and AutoRoute showed less accuracy in meandering channels and near confluences. Overall, the AutoRoute model produced slightly higher flood extent accuracy than the HAND model. However, the AutoRoute model tended to have lower accuracy with lower flow events. This highlights a concern that

the AutoRoute model has typically been tested for large flood events (flood events greater than the 50-yr flood were tested in Follum (2012), Follum et al. (2017; 2018; 2019), and Wahl et al. (2017)) and may not be applicable for less extreme flow events.

This article presents modifications to the AutoRoute model to better incorporate bathymetry estimations and terrain in the

calculation and post-processing of flood inundation maps, which are expected to improve the flood mapping capability of the AutoRoute model for extreme (>50 yr flood event) and non-extreme flood cases. The modifications are expected to produce continuous and accurate flood extent results for both low and high flow events. The AutoRoute model is tested at seven locations within the U.S. where flood inundation maps for multiple flow rate scenarios (ranging from low to high flow events) have been modelled and compared to observed flow events by NOAA's Advanced Hydrologic Prediction Service (McEnery

et al., 2005).

## 2 Methodology

### 2.1 AutoRoute Model

AutoRoute is a grid-based model where elevation, stream locations (stream cells), and land cover are defined using a raster format. Gridded stream cells were originally defined using a flow accumulation raster (Follum, 2012). With the creation of river networks in polyline format (e.g. NHDPlus and HydroSHEDS) stream cells are now created by converting polyline data to a raster or table format (table defines the x- and y- coordinates). Each stream cell retains the unique river reach identifier (e.g. ComID in NHDPlus) to associate attributes of the stream reach to each stream cell. For example, streamflow $Q$ (m$^3$ s$^{-1}$) from a hydrologic model, such as SPT or NWM, is assigned to each stream cell using the river reach identifier. At each stream cell, cross-sections are sampled from an elevation dataset (Figure 1). In the original AutoRoute model the channel profile is estimated only from the elevation dataset; no bathymetric profile is assumed. Although not assuming a bathymetric profile was acceptable in the original applications of AutoRoute where ephemeral streams were being simulated, AutoRoute is being used in more regions and inclusion of bathymetric profiles should improve flood inundation estimations (Dey et al., 2019) and mobility assessments. For each cross-section sampled AutoRoute now includes a bathymetry estimation. AutoRoute adjusts the centerline of the cross-section to the lowest point. The lateral distance that AutoRoute searches for the lowest point is specified by the user, typically defined as 20m. As shown in Figure 1, the cross-section sampled from the DEM often shows the stream/river as a flat surface. AutoRoute automatically finds the top-width ($T$, m) of the water surface and then estimates a bathymetric profile. The bathymetric profile is assumed to have an exponential shape, as shown in Figure 1. The exponential shape takes the form

$$y = \delta \, |x|^{\beta}, \tag{1}$$

where $y$ (m) and $x$ (m) are the ordinates of the bathymetric profile as shown in Figure 1 and $\delta$ is a user-defined parameter (assumed 0.001 for this paper). When $x=0$ the maximum depth of the bathymetric profile ($Z$, m) is assumed to occur ($y = 0$ based on the orientation of the y and x axes in Figure 1). When $x = T/2$ the bathymetric profile is at the bank of the river and $y = Z$. Based on these two constraints and Eq 1, $\beta$ is calculated as

$$\beta = \frac{log(Z/\delta)}{log(T/2)}. \tag{2}$$

Using Manning's equation (described below), $Z$ is calculated so that a specified base flow will pass through the bathymetric profile. The bathymetric profile is burned into the cross-section profile and the centerline of the stream/river is again adjusted to the lowest point. Hydraulic area $A$ (m$^2$) and wetted perimeter $P$ (m) are calculated at each cross-section for a given flow depth $D$ (m). Using a volume-fill approach $D$ is incrementally increased until there is less than a 1% difference between $Q$ and the calculated streamflow $Q_{calc}$ (m$^3$ s$^{-1}$), calculated using Manning's Equation:

$$Q_{calc} = \frac{c_u}{n} A^{5/3} P^{-2/3} S_f^{1/2}, \tag{3}$$

where $c_u$ is the unit constant (1.0 for metric units), $n$ is the Manning's roughness coefficient, and $S_f$ is the hydraulic slope. Normal depth is assumed, and therefore $S_f = S_o$, where $S_o$ is the slope of the channel. AutoRoute calculates $S_o$ by analyzing

the elevations and lateral distances upstream and downstream of the stream cell being analyzed (more explanation found in Follum et al. (2017)). $n$ is estimated as (Horton, 1933; Einstein, 1934):

$$n = \left[ \sum_{i=1}^{N} \frac{P_i n_i^{1.5}}{P} \right]^{2/3},$$
(4)

where $P_i$ and $n_i$ are wetted perimeter and Manning roughness coefficient of the $i$th segment within the cross-section, and $N$ is the total number of segments within the cross-section that are flooded. $n_i$ values are associated with land cover types, as described in Follum et al. (2017).

An initial cross-section is sampled perpendicular to the stream direction, as defined by positions of upstream and downstream stream cells. However, stream cross-sections may not always adequately capture the floodplain geometry, therefore multiple cross-sections are sampled for each stream cell by incrementally pivoting the cross-section relative to the stream direction. As shown in Follum et al. (2017), these multiple cross-sections have the effect of filling in the floodplain but can also create errant cross-sections and therefore errors in the floodplain mapping. The cross-section for each stream cell (subscript $sc$) that

produces the shortest top width $TW_{sc}$ (m) is expected to be the most representative cross-section for that stream cell. The $TW_{sc}$ and the flow depth $D_{sc}$ (m) of the representative cross-section are recorded for each stream cell.

AutoRoute originally created flood inundation and flood depth rasters by mapping all of the cross-section depths and extents onto a raster (Follum 2012). Later, an iterative combination of the Boundary Clean and Aggregate Polygons functions within

ArcGIS (ESRI, 2011) was then used to fill-in holes, omit outlier flood cells, and smooth boundaries along the flood polygon (Follum et al., 2017). None of the previous post-processing considered topography in the creation of the flood polygon. In this paper the use of Boundary Clean and Aggregate Polygons functions within ArcGIS (ESRI, 2011) are considered as the baseline method for post-processing AutoRoute results and is referred to as GIS Post-Processing (GISPP).

**2.2 Development of AutoRoute post-processing script (ARPP)**

The AutoRoute post-processing script (ARPP) has been developed to better account for topography when creating the flood inundation map. The water surface elevation of each stream cell $WSE_{sc}$ (m) is calculated:
$$WSE_{sc} = E_c + D_{sc},$$
(5)
where $E_c$ (m) is the elevation of the cell. The water surface elevation for each cell in the model domain ($WSE_c$, m) is interpolated from the $WSE_{sc}$ values using inverse-distance-weighting:

$$WSE_c = \frac{\sum WSE_{sc} w}{\sum w},$$
(6)
where $w$ is the weight, calculated as:
$$\begin{cases} w = d_{c \to sc}^{-2} & if \quad d_{c \to sc} \leq \alpha \, TW_{sc} \\ w = 0 & if \quad d_{c \to sc} > \alpha \, TW_{sc} \end{cases}$$
(7)

where $d_{c \to sc}$ (m) is the distance between the model domain cell and the stream cell, and $\alpha$ is a user-defined parameter. Higher values of $\alpha$ increase the influence that each stream cell has on flooding the surrounding cells. The flood depth for each cell in the domain $D_c$ (m) is then calculated as:

$$D_c = WSE_c - E_c, \tag{8}$$

where $D_c$ values less than zero are set to zero and cells with $D_c$ values greater than zero are considered flooded. All flooded cells are then converted to a polygon format.

Figure 2 (top) demonstrates the flooding ($D_c$ values) of the surrounding terrain from a single stream cell. When the depths from all stream cells are included by use of Eqs. 6 and 7 the flooding of the surrounding cells provides a continuous flood map with holes only in the high-elevation areas (bottom frame of Figure 2). Additionally, stream cells that have $WSE_{sc}$ values higher/lower than surrounding stream cells (i.e. outliers) have impact only on the immediately surrounding cells (see shallow locations within river in bottom frame of Figure 2). These outliers can be caused by cross-sections not being perpendicular to the stream reach, errors in hydraulic slope estimation, and errors within the DEM. Although these outliers affect the immediately surrounding cells, they have minimal impact on flooding in the floodplain. However, these outliers could affect channel profiles for mobility analysis and should be addressed in future research. The minimal impact of outliers on flood inundation is due to the influence of water surface elevations from multiple stream cells on each $WSE_c$ value. Use of ARPP to post-process AutoRoute flood depth results is expected to produce more continuous flood maps, account for topography, and reduce the impact of errant $D_{sc}$ values on the flood inundation results, all of which are expected to be important in simulating both low- and high-flow events.

## 2.3 Study Locations

For several communities throughout the United States the USGS has created flood inundation maps for multiple water surface elevations (stages) of the river. These maps are intended to be used in conjunction with National Weather Service (NWS) forecasted peak-stage data to show predicted areas of flooding. The modelled stage heights vary between the sites but are intended to capture the river stage at multiple (often around 20) stages between normal conditions (low flow) and the highest rated stage at the streamgage (high flow). The hydraulic model used to create the flood inundation maps varies between the sites, but each model is validated against observed flood events. For this study seven locations where the U.S. Geological Survey (USGS) has completed flood inundation studies were chosen (Figure 3). Each site varies in complexity as well as geographical location (multiple river basins throughout the U.S.).

For each site used in this study Table 1 lists the location, identification (ID), river(s), USGS streamgage number, length of river segments within the study, and reference. All studies utilized LiDAR elevation datasets ranging between 0.9 and 3 m horizontal spatial resolution. The HEC-RAS hydraulic model (USACE, 2010; 2016) was used in each study and was calibrated and validated to observed flood data.

Table 2 lists the base flow and the low, medium, and high flow rates used in the study. The low, medium, and high flow rates were chosen based on the minimum, median, and maximum modelled flow rates in each of the USGS studies (a flow rate was assigned to each stage height in each of the studies). The USGS does not provide base flow estimates for the sites in this study, so the base flow was estimated as the average annual flow rate for each gage listed in Table 1. The annual flow rates were obtained from USGS WaterWatch (https://waterwatch.usgs.gov/?id=ww_current; visited 01 Feb 2019). USGS streamgage 02126375 along the Pee Dee River (Figure 8) does not record flow rates, so the flow data from the USGS streamgage 0212378405 approximately 12-km upstream along the Pee Dee River was used to estimate baseflow. Brown Creek and Rocky River are also included in the NC study (Smith and Wagner, 2016), but are omitted from this study because flow rates were unavailable. The USGS streamgage 02473000 along the Leaf River is used in the MS study and is less than 1 km downstream of the confluence of the Leaf and Bouie Rivers (Figure 9). Above the confluence of the rivers the Leaf and Bouie Rivers are assumed to carry approximately 70% and 30%, respectively, of the flow rates measured at the USGS streamgage 02473000 (Storm, 2014).

## 3 Model Application

AutoRoute models were developed for each of the seven test locations. Each model was developed using elevation data from the 1/3-arc-second (~9 m) National Elevation Dataset (Gesch et al., 2002), and land cover classifications were obtained from the 2011 National Land Cover Database (NLCD) (Homer et al., 2015). The NLCD has a spatial resolution of approximately 30 m and therefore was resampled to the resolution of the DEM. The stream networks for each study site were defined using the NHDPlus dataset.

For each simulation, the qualitative performance of the AutoRoute models compared to the USGS data are measured using the F-statistic ($F$, percentage) (Bates and De Roo, 2000; Tayefi et al., 2007) and error bias ($E$) (Wing et al. 2017):

$$F = 100 \left( \frac{A_{Acc}}{A_{Obs} + A_{Sim} - A_{Acc}} \right), \tag{9}$$

$$E = \frac{A_{Over}}{A_{Under}}, \tag{10}$$

where $A_{Obs}$ (km$^2$) is the area of flooding from the USGS flood maps, $A_{Sim}$ (km$^2$) is the area of flooding from the AutoRoute simulation, $A_{Acc}$ (km$^2$) is the area where both AutoRoute and the USGS show flooding, $A_{Over}$ (km$^2$) is the area where only the AutoRoute model shows flooding, and $A_{Under}$ (km$^2$) is the area where only the USGS flood maps shows flooding. $F$ ranges between 0 and 100% with a value of 100% indicating perfect fit between the AutoRoute and USGS flood inundation maps. Previous applications of AutoRoute within the U.S. have had $F$ values between 58.4 and 92.5% (Follum et al., 2017), with the IN test site having an $F$ value of 77% when compared to observed flood maps from the June 2008 flood. $E$ ranges between 0

and ∞ with $E$ values less than 1indicating a bias towards underestimation, $E$ values greater than 1indicating a bias towards overestimation, and an $E$ value of 1 indicating no bias.

The AutoRoute model has few calibration parameters. Following Follum et al. (2017), $n_i$ values were set to the lower bound
as described in Moore (2011), Chow (1959), and Calenda et al. (2005). The number of cross-sections sampled at each stream cell was set to 9 following Follum et al. (2017). The influence that each stream cell has on flooding the surrounding cells is controlled by the user-defined $\alpha$ parameter. A sensitivity analysis was performed to determine the affect $\alpha$ has on the flood inundation when using ARPP post-processing. Using all seven test sites and all three flow scenarios $\alpha$ was varied between 0.25 and 3.0 by increments of 0.25. Figure 4 shows the $F$ value associated with each of the 252 simulations. Increases in $\alpha$
tend to result in an increase in accuracy (higher $F$ values). However, increases in $\alpha$ also increase the computational burden. For example, four times as many cells are analysed for each stream cell when $\alpha$=3.0 than when $\alpha$=1.5. For this study $\alpha$ is set to 1.5 because it provides good coverage of the river floodplain (and thus higher $F$ values) while remaining computationally efficient.

## 4 Results and Discussion

### 4.1 Flood Inundation Mapping

For each study site the low, medium, and high flow scenarios were simulated using AutoRoute. The results were then post-processed using the ARPP method (AutoRoute+ARPP) described in this paper as well as the original GISPP method (AutoRoute+GISPP). The only difference between AutoRoute+GISPP and AutoRoute+ARPP results is the post-processing method used. For each test case Table 3 shows the quantitative performance ($F$ and $E$) of flood inundation maps simulated
using AutoRoute+ARPP and AutoRoute+GISPP as compared to the USGS flood inundation maps. Table 3 also shows the value of $A_{Obs}$, $A_{Sim}$, $A_{Over}$, $A_{Under}$, and $A_{Acc}$ for each flood inundation map simulated using AutoRoute+ARPP. Overall, the use of ARPP results in improved flood inundation accuracy when compared to GISPP (average $F$ value is 7.4% higher when using ARPP). The increase in accuracy of the ARPP method is most evident in the low-flow scenarios where the average $F$ value increases from 45.8% when using GISPP to 63.3% when using ARPP. For the med-flow scenarios the average $F$
value increases from 65.7% when using GISPP to 70.0% when using ARPP. The difference in $F$ value for the high-flow scenario is minimal when using GISPP (77.2%) and ARPP (77.5%). Flooding results are split between overestimation ($E$>1) and underestimation ($E$<1) when using the ARPP method (10 test cases underestimated and 11 test cases overestimated), while the GISPP method is more prone to overestimation (6 test cases underestimated and 15 test cases overestimated). Overall, the use of ARPP improved the accuracy (higher $F$ value) of the flood results in 18 of the 21 test cases.

Figures 5-7 show a comparison between flood inundation maps generated using AutoRoute+ARPP and the USGS flood maps. In the figures the areas shaded green (Accurate) indicate areas where AutoRoute+ARPP and the USGS flood maps agree. Areas shaded red (Over) indicate where only AutoRoute+ARPP simulates the area as flooded and areas shaded blue (Under) indicate where only the USGS shows the area as flooded. The accuracy of the flood maps generated using low flows (average $F$ value of 63.3%) are comparable with results from other studies (Afshari et al., 2017; Dey et al., 2019; Follum et al., 2017; Tayefi et al., 2007), but tend to overestimate flooding (all $E$ values are greater than 1 except for the CO test site). Although IN has the highest $E$ value, the high $F$ value and Figure 5 show the flood map during the low flow event is accurately simulated and the $E$ value is inflated due to the minimal underestimation of flooding (Table 3). Visually and quantitatively, NC and MS (Figure 5 and Table 3) have the greatest amount of overestimation during the low flow event, resulting in the lowest $F$ values of all the simulations. NC shows overestimation in low-lying areas adjacent to the river where the ARPP allows for flooding in areas even if they are not hydraulically connected to the streamlines, resulting in the lowest overall $F$ value of 39.3%. MS also shows gross overestimation of flooding during the low-flow event. MS has minimal topography, a characteristic that has shown AutoRoute to produce less accurate results (Follum et al., 2017). AutoRoute simulations are essentially one-dimensional (1D); better representation of hydrodynamics in areas with minimal topography occurs with multi-dimensional modelling. Additionally, MS has the highest ratio of low flow to base flow (the low flow used in this study is over 15 times the flow rate of the base flow) which may have led to errors in bathymetry estimation if the elevation dataset was derived during a higher flow event. The coarse resolution used in this study compared to the USGS study may also contribute to inaccuracies (e.g. overestimation) that may be more pronounced in flatter terrain such as MS. While most streams considered in this analysis lie in rural land use environments, such as forested or agricultural areas, MS occurs in a primarily urban to sub-urban environment where small-scale changes in the topography are smoothed or negated in the relatively coarse ~9 m DEM. Many of these missed topographic features are likely flood control structures, such as levees. The combination of minimal topography, DEM inaccuracies, and land use complexities likely led to the overestimation found in the MS study.

With a few exceptions (e.g. SC), the flood maps generated for the med-flow events ($F$ value of 70.0%; Figure 6) and high-flow events ($F$ value of 77.5%; Figures 7) are more accurate than the flood maps generated for the low-flow events($F$ value of 63.3%; Figure 5). This finding is consistent with flood mapping results using the HEC-RAS hydraulic model in recently published work (Dey et al., 2019). They also showed that the median of F values is higher with increasing flow. The maximum $F$ value of 92.6% occurs at NC during the med flow (NC had the lowest overall $F$ value during the low-flow event). The sudden increase in $F$ value between the flood maps generated using low-flow and med-flow at NC is due to the low-lying terrain near the river being simulated as flooded by both AutoRoute+ARPP and the USGS during the med-flow event, thus reducing the overestimation and increasing the accuracy. Although flood maps for the med- and high-flow events tend to have higher $F$ values, they also tend to have a bias to underestimate the flooded area ($E$ values less than 1). The majority of underestimation at the IN test site (Figures 6 and 7) occurs where a tributary (Meadowbrook Creek) that is not accounted for in the AutoRoute simulation flows into the White River to the south and west of the town of Spencer.

The two test locations along the Deerfield River in Massachusetts (MC and MW) show consistent accuracy between the low-, med-, and high-flow rates. This region of Massachusetts has well-defined rivers and medium to high topographic relief. These features allow AutoRoute to better capture the riverbanks and floodplain, resulting in consistent accuracy ($F$ values close to 100) and minimal bias ($E$ values close to 1).

## 4.2 Flood Inundation Mapping Test Using High-Resolution DEM

Elevation datasets are used in flood mapping to define the topographic features (slopes, banks, levees, etc.) of the area being modelled, therefore the spatial resolution and vertical accuracy of the elevation datasets being used have a large impact on the accuracy of flood inundation maps being generated (Ali, Solomatine, and Di Baldassarre, 2015; Brandt and Lim, 2012; Cook and Merwade, 2009; Hsu et al., 2016). A thorough investigation of the impacts of various elevation datasets on the accuracy of flood inundation maps generated using AutoRoute+ARPP is outside the scope of this paper. However, a simple test is employed to determine if a high-resolution DEM (~3 m) improves the flood inundation accuracy when using AutoRoute+ARPP. The MS and NC sites had the most overestimation of flooding during the low flow event when using a ~9 m resolution DEM and are therefore used in this test. A 1/9-arc-second (~3 m) National Elevation Dataset (Gesch et al., 2002) elevation dataset replaces the ~9 m elevation dataset. The NLCD was resampled to 1/9 arc-second, but all other data remains the same from the previous tests. Figures 8 and 9 show the high-resolution flood inundation for low-, med-, and high-flow events at NC (NC-3m) and MS (MS-3m), respectively. Table 3 shows the quantitative performance for flood inundation maps simulated using AutoRoute+ARPP compared to the USGS flood inundation maps. Table 3 and Figures 5-8 show the flood results for NC and NC-3m are similar for each of the flow events. Even with the high-resolution DEM the model still simulates flooding in the low-lying terrain near the river in the low-flow event (Figures 5 and 8), thus resulting in a high overestimation (high $E$ and $A_{Over}$ values in Table 3). Comparing the simulated area ($A_{Sim}$) for the MS test site the model using 3-m DEM data produced a flood map having approximately half the area of the flood map using a ~9 m DEM (Figures 5 and 9). For the med- and high-flow test cases the $A_{Sim}$ values were approximately 78% and 97%, respectively, of the values when using a ~9 m DEM. Use of higher resolution at the MS test site produced smaller $A_{Sim}$ values (Table 3 and Figure 9), especially for smaller flood events. Although smaller $A_{Sim}$ values resulted in lower $A_{Over}$ values for MS-3m, they also resulted in higher $A_{Under}$ values which showed a bias of the model to underestimate the flooded area and therefore the $F$ values did not improve when compared to the MS results. In general, the higher resolution DEM did not substantially improve flood inundation results in NC or MS as expected.

Regardless of DEM resolution, inaccuracies in flood inundation results may be due to the use of constant Manning's roughness coefficient values ($n_i$) that are set solely based on land cover maps. Not only are roughness coefficients likely different even under the same land cover types, but the values of $n_i$ also vary with the depth of water (Ree and Palmer, 1949; Temple et al., 1987). In this study the low estimate of $n_i$ values were used based on Follum et al. (2017). However, that study did not include

bathymetry estimation within the cross-sections and therefore a reexamination of the proper of $n_i$ values to use within AutoRoute may be warranted. Another source of error may be the simple bathymetry estimation for each cross-section. A more detailed bathymetry would affect the low-flow scenario the most but would likely improve the accuracy of flood inundation for all flow scenarios.

## 4.3 Simulation Time

On average, each flow event for each test case using the ~9 m DEM took approximately 12 seconds to read all data (elevation, land cover, stream location, and flow rates) into memory, simulate flood depth results using AutoRoute, post-process the flood depth results into raster flood maps using ARPP, and convert the raster flood maps into flood inundation polygons. For the high-resolution test locations (MS-3m and NC-3m) each test case took over 90 seconds. However, all of the test cases were for relatively small areas whereas the main reason to utilize a simplified hydraulics model such as AutoRoute is for computational efficiency when simulating flood inundation along thousands of river reaches at the regional- to continental-scale. Therefore, to compare computation times to the original AutoRoute methods described in Follum et al. (2017) the same domains in the Midwest (230,000 km$^2$ area) and Mississippi Delta (109,500 km$^2$ area) were simulated again using the methods described in this paper (AutoRoute+ARPP). Similar to Follum et al. (2017), the domains were discretized into thirty-nine 1° by 1° tiles (as defined by how USGS NED data is disseminated). Flow rates from Tavakoly et al. (2017) were once again used to define the peak flow in each river reach in the domain. The AutoRoute simulations in Follum et al. (2017) required approximately 20-minutes to simulate a 1° by 1° tile, compared to 17.5-minutes using the current version of AutoRoute. The current version of AutoRoute is more computationally efficient through the use of the Geospatial Data Abstraction Library (GDAL/OGR Contributors, 2019) for reading and writing data. The post-processing procedure (i.e. GISPP) described in Follum et al. (2017) required approximately 15-minutes for each 1° by 1° tile. Post-processing using ARPP to convert flood depth data to a flood depth raster and flood polygon takes approximately 3 minutes. Overall, the current version of AutoRoute and the use of ARPP is over 40% more computationally efficient in simulating flood inundation maps.

The increased computational efficiency of AutoRoute and ARPP along with removing the requirement for ArcGIS software in post-processing may allow for the AutoRoute model to more effectively be implemented on computational servers by CHL to provide flood and mobility assessments for OCONUS applications. These assessments will likely use SPT for streamflow data and be operated at the regional-scale using a 1° by 1° spatial discretization. A further modification to improve computational efficiency may be to create a database of AutoRoute simulations for varying flow rates. When forecast flowrates become available the database could be used instead of an AutoRoute simulation to determine the depth within each stream cell. ARPP could then be used to generate the flood maps. This type of database could also provide flow-depth relationships to be used with the HAND method. Additionally, a production system could determine if streams within each modelling domain cross a specified bankfull streamflow threshold and AutoRoute simulations would only occur if the streamflows for a

350 given hydrometeorlogical forecast exceeded these bankfull thresholds. Either process may further improve the computational efficiency in creating production flood inundation maps.

## 5 Conclusions

The AutoRoute model is a simplified hydraulics model designed to quickly provide high-resolution flood inundation and mobility results at the regional to continental scale. The main purpose of this paper was to test the computational efficiency

and accuracy of flood inundation maps generated by the AutoRoute model with special consideration given to less-extreme flow events (i.e. low and medium flood events). Seven test sites were chosen to compare flood inundation maps using low-, medium-, and high-flow rates. The seven test sites used a ~9 m elevation dataset and the locations correspond to existing USGS flood inundation studies and represent different regions within the U.S. The primary conclusions of the paper are as follows:

1.) Implementation of a new post-processing procedure improved the flood inundation accuracy of the AutoRoute model, especially when simulating low-flow events (average $F$ value increase of 17.5% when compared to previous post-processing methods). Although the flood inundation results for low-flow events are comparable with other studies (average $F$ value of 63.3%), the simulated flooding tends to be overestimated. Higher-flow scenarios tend to be more accurately simulated ($F$ value for medium-flow events is 70.0% and average $F$ value for high-flow

events is 77.5%). Simplifications in estimating roughness coefficients, cross-section profiles (including bathymetry estimation), and the hydraulic simulation allow for AutoRoute to be computationally efficient but also may lead to errors in flood map simulation.

2.) Recent updates to the input and output methods within AutoRoute model as well as the post-processing procedure allow for the creation of flood inundation rasters (~9 m resolution) and polygons in 20.5 minutes for a 1° by 1° area,

as compared to 35-minutes in previous studies. Increased computational efficiency may allow for the AutoRoute model to more effectively be implemented in a production environment at the regional to continental scale.

3.) Use of higher-resolution (~3 m) elevation data within the AutoRoute model was also tested at two of the sites and did not significantly improve the accuracy of the flood inundation maps. One of the sites showed only minimal difference in flood inundation when using the higher-resolution elevation data. The second site had an almost 50%

reduction in simulated area for the low-flow test case, which reduced the overestimation of flooded area but also increased the underestimation of flooded area. Use of the higher-resolution elevation datasets increased computation time by 750% compared to when the ~9 m elevation dataset was used.

4.) As has been found in other studies, AutoRoute performs best in areas with mid-to-high topographic relief where one-dimensional flood models often perform well. Areas of minimal relief are more susceptible to back-water

effects. AutoRoute physics do not account for such physical complexities and model results tend to be less accurate.

As such, flood inundation results from AutoRoute should be viewed as a first-order approximation with the use of more detailed hydraulic models providing more actionable flood data.

The scope of this research was limited to small and medium inland rivers within the U.S. Several areas of future research were highlighted, including the need to better estimate roughness coefficients based on land cover and to account for change in roughness with flow depth. Improved elevation data and bathymetry estimation could increase the accuracy of both the flood inundation estimates and mobility assessment when using AutoRoute. Based on the recent work by Dey et al. (2019), different bathymetric methods could be implemented into AutoRoute for differing geomorphological conditions. Removal of outlier flood depth values will also improve the flood inundation estimation as well as the channel profiles that are used for mobility analysis. Use of a database system to store precomputed AutoRoute results could also increase computational efficiency and connect to other hydraulic models, such as HAND. Flood inundation models capable of quickly providing high-resolution flood maps have seen significant development over the past decade as regional- to continental-scale flow simulation models are becoming operationalized by the U.S. Army, NOAA, and others. While the flow and flood inundation models continue to advance, the connection between the flood maps generated and the impacts to the population/environment need to become more fully-developed.

**Acknowledgements**

This project was supported by the Deputy Assistant Secretary of the Army for Research and Technology through the Engineer Research and Development Center's Military Engineering applied research work package title Austere Entry; the Geospatial Intelligence Directorate of the Marine Corps Intelligence Activity; and ERDC's collaborative research project supporting the NOAA National Water Center. We would also like to acknowledge the anonymous reviewers for their insightful comments that helped improve this article.

**Code / Data Availability**

All elevation, land cover, streamflow, and USGS flood inundation data used in this paper are publicly available (see references throughout paper as well as Table 1). All output data and information regarding the AutoRoute model and AutoRoute post-processing script (ARPP) are available by contacting the corresponding author.

**Author Contribution**

All authors contributed to this article, with the order of the authors' names reflecting the size of their contribution.

**Competing Interests**

The authors declare that they have no conflict of interest.

**Special Issue Statement**

This article is part of the special issue "Advances in Computational Modelling of Natural Hazards and Geohazards".

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

**Tables and Figures**

**Table 1: USGS study sites used in this study.  For each study site the location, ID, river(s), USGS streamgage, model length, and reference are provided.**

| Location | ID | River(s) | USGS Streamgage Number | Model Length (km) | Reference |
|---|---|---|---|---|---|
| Spencer, IN | IN | White River | 03357000 | 8.5 | Nystrom (2013) |
| Fort Morgan, CO | CO | S. Platte River | 06759500 | 7.2 | Kohn and Patton (2018) |
| Greenville, SC | SC | Saluda | 02162500 | 6.4 | Benedict et al. (2013) |
| Pee Dee, NC | NC | Pee Dee River | 02126375 | 17.0 | Smith and Wagner (2016) |
| Hattiesburg, MS | MS | Leaf and Bouie Rivers | 02473000 | 10.9 | Storm (2014) |
| Charlemont, MA | MC | Deerfield River | 01168500 | 14.6 | Lombard and Bent (2015) |
| West Deerfield, MA | MW | Deerfield River | 01170000 | 14.3 | Lombard and Bent (2015) |

**Table 2: Base flow and Low, Medium (Med), and High flow rates for each study site.**

| ID | Base Flow (m³ s⁻¹) | Low Flow (m³ s⁻¹) | Med Flow (m³ s⁻¹) | High Flow (m³ s⁻¹) |
|----|----------|----------|----------|----------|
| IN | 83.9 | 164.0 | 577.7 | 2027.5 |
| CO | 16.6 | 79.9 | 577.7 | 2814.7 |
| SC | 17.2 | 79.6 | 222.9 | 373.8 |
| NC | 145.4 | 911.8 | 3021.1 | 7391.8 |
| MS | 63.5 | 999.6 | 1730.2 | 3409.3 |
| MC | 25.9 | 311.5 | 996.8 | 2415.4 |
| MW | 38.2 | 455.9 | 1659.4 | 3344.2 |

**Table 3: F-statistic (*F*, percentage) and error bias (*E*) for AutoRoute+ARPP and AutoRoute+GISPP for each flow scenario at all seven test location. Inundation coverage areas ($A_{Obs}$, $A_{Sim}$, $A_{Over}$, $A_{Under}$, and $A_{Acc}$) are also shown for AutoRoute+ARPP. *F* ranges between 0 and 100% with a value of 100% indicating perfect fit between the simulated and USGS flood inundation maps. *E* values less than 1 indicate a bias towards underestimation, *E* values greater than 1 indicate a bias towards overestimation, and an *E* value of 1 indicates no bias. All inundation coverage areas have units of km². *F*, *E*, and inundation coverage areas are also shown for the two sites tested using higher-resolution elevation data (MS-3m and NC-3m).**

| Location | Flow Rate | AutoRoute + ARPP | | | | | | | AutoRoute + GISPP | |
|---|---|---|---|---|---|---|---|---|---|---|
| | | $A_{Obs}$ | $A_{Sim}$ | $A_{Over}$ | $A_{Under}$ | $A_{Acc}$ | *F* | *E* | *F* | E |
| IN | Low | 0.75 | 0.91 | 0.18 | 0.02 | 0.72 | 78.2 | 8.22 | 54.9 | 37.74 |
| | Med | 2.36 | 2.89 | 0.86 | 0.33 | 2.03 | 63.0 | 2.60 | 52.5 | 5.81 |
| | High | 5.48 | 4.66 | 0.10 | 0.92 | 4.56 | 81.7 | 0.11 | 78.4 | 0.54 |
| CO | Low | 0.71 | 0.66 | 0.13 | 0.18 | 0.53 | 63.1 | 0.70 | 53.1 | 3.19 |
| | Med | 2.68 | 1.89 | 0.14 | 0.93 | 1.75 | 62.2 | 0.15 | 59.0 | 2.04 |
| | High | 6.67 | 5.87 | 0.17 | 0.97 | 5.69 | 83.2 | 0.18 | 80.9 | 0.40 |
| SC | Low | 0.26 | 0.30 | 0.07 | 0.03 | 0.23 | 71.6 | 2.59 | 42.7 | 53.14 |
| | Med | 0.63 | 0.51 | 0.08 | 0.20 | 0.43 | 60.6 | 0.39 | 59.1 | 4.33 |
| | High | 0.97 | 0.70 | 0.04 | 0.31 | 0.66 | 65.9 | 0.12 | 75.8 | 1.41 |
| NC | Low | 3.65 | 7.82 | 4.58 | 0.42 | 3.23 | 39.3 | 10.97 | 27.7 | 15.43 |
| | Med | 20.27 | 20.48 | 0.89 | 0.67 | 19.60 | 92.6 | 1.32 | 87.0 | 1.55 |
| | High | 25.69 | 23.34 | 0.43 | 2.78 | 22.91 | 87.7 | 0.15 | 86.5 | 0.31 |
| MS | Low | 6.45 | 9.78 | 4.66 | 1.33 | 5.12 | 46.1 | 3.49 | 42.2 | 4.91 |
| | Med | 14.40 | 12.59 | 1.71 | 3.52 | 10.88 | 67.6 | 0.49 | 65.9 | 0.80 |
| | High | 21.46 | 16.98 | 1.13 | 5.61 | 15.85 | 70.2 | 0.20 | 68.9 | 0.60 |
| MC | Low | 1.26 | 1.39 | 0.21 | 0.08 | 1.18 | 80.0 | 2.62 | 57.3 | 24.96 |
| | Med | 2.18 | 2.39 | 0.41 | 0.19 | 1.99 | 76.7 | 2.10 | 57.9 | 6.31 |
| | High | 3.42 | 3.73 | 0.55 | 0.25 | 3.18 | 80.0 | 2.25 | 67.1 | 6.95 |
| MW | Low | 1.76 | 1.77 | 0.38 | 0.37 | 1.39 | 65.0 | 1.02 | 43.0 | 8.97 |
| | Med | 6.91 | 4.97 | 0.19 | 2.12 | 4.79 | 67.5 | 0.09 | 78.3 | 1.14 |
| | High | 8.57 | 6.93 | 0.34 | 1.98 | 6.59 | 74.0 | 0.17 | 83.0 | 0.99 |
| MS-3m | Low | 6.45 | 5.18 | 1.56 | 2.83 | 3.62 | 45.2 | 0.55 | | |
| | Med | 14.40 | 9.90 | 1.10 | 5.59 | 8.81 | 56.8 | 0.20 | | |
| | High | 21.46 | 16.41 | 0.66 | 5.71 | 15.75 | 71.2 | 0.12 | | |
| NC-3m | Low | 3.65 | 7.75 | 4.53 | 0.42 | 3.23 | 39.5 | 10.69 | | |
| | Med | 20.27 | 20.68 | 0.90 | 0.48 | 19.79 | 93.5 | 1.86 | | |
| | High | 25.69 | 23.39 | 0.42 | 2.72 | 22.97 | 88.0 | 0.15 | | |

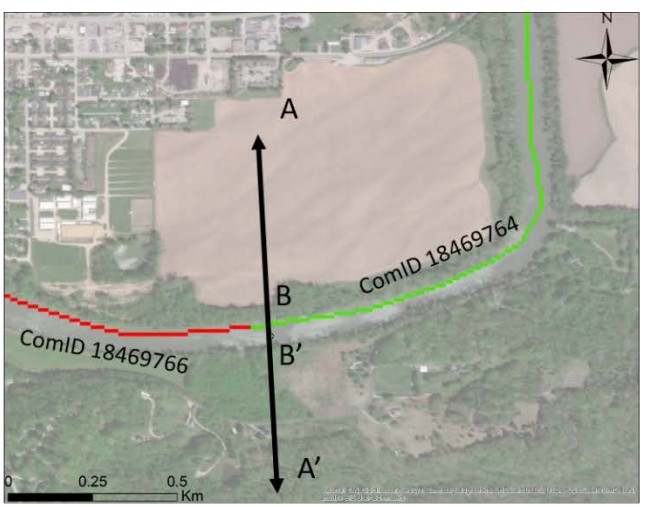

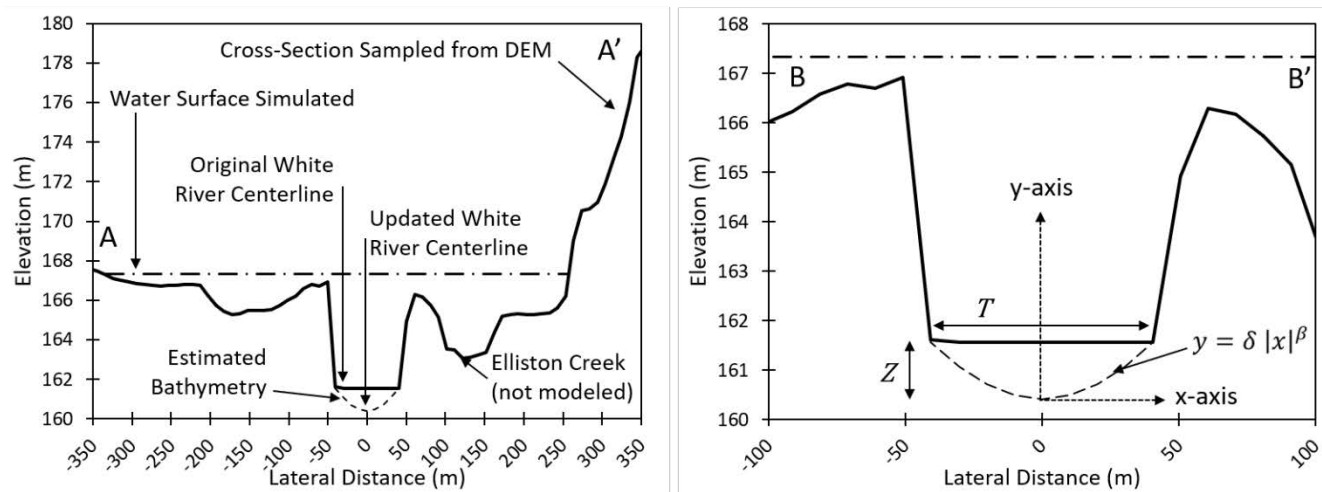


**Figure 1: Cross-section profile of the White River near Spencer, IN. Also shown is the bathymetry estimation where $T$ is the top-width of the channel, $Z$ is the maximum depth of the bathymetry profile, and $y$ and $x$ are the ordinates of the bathymetry profile. Sources of the background imagery in Figures 1-3 and 5-9 include ESRI, DigitalGlobe, Earthstar Geographics, CNES/Airbus DS, GeoEye, USDA FSA, USGS, Getmapping, Aerogrid, IGN, IGP, and the GIS User Community.**


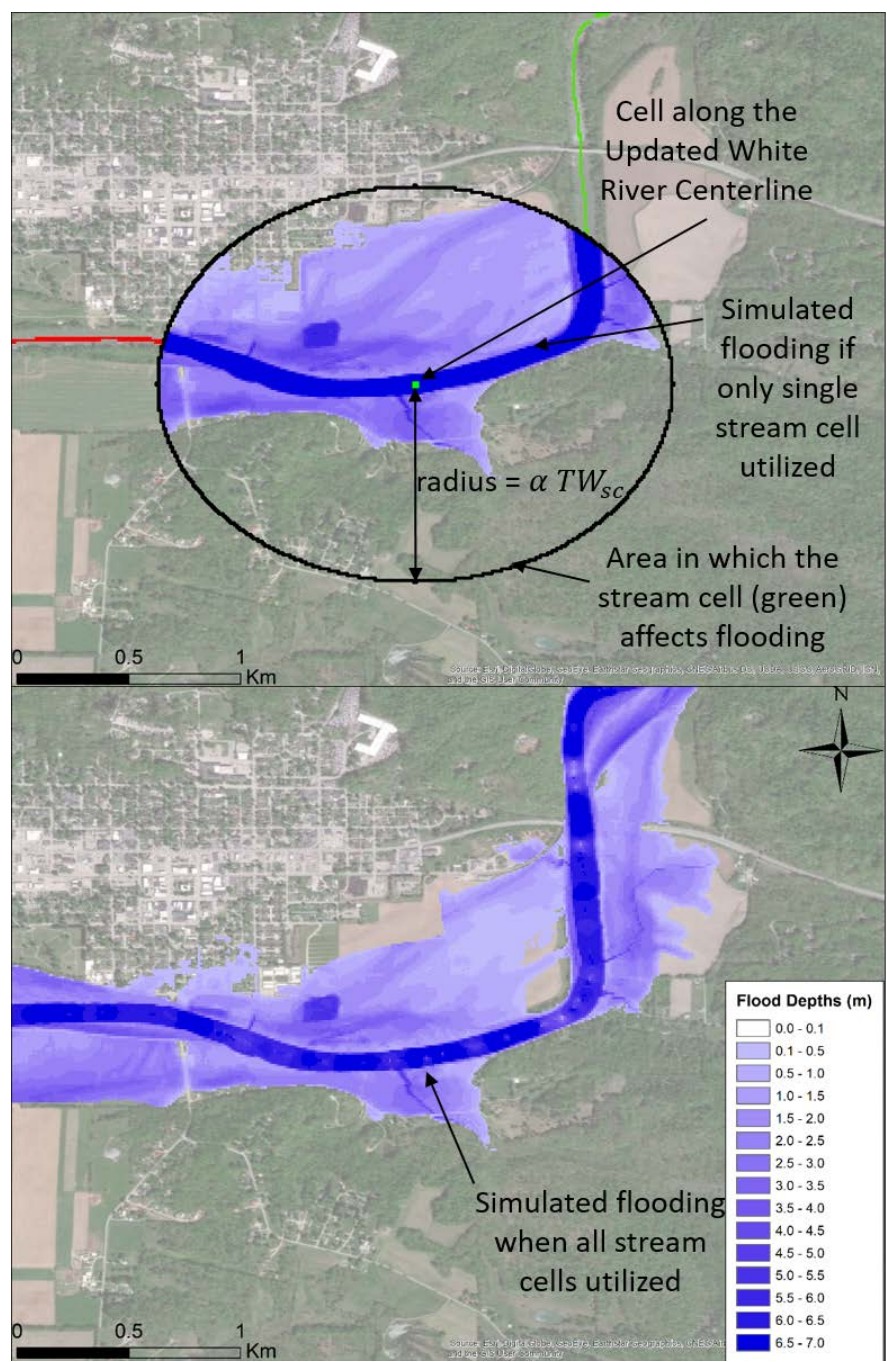

**Figure 2: Top shows flood depths of surrounding terrain from at a single stream cell. Notice the area of influence (cells within radius=$\alpha \, TW_{sc}$) appears elliptical due to projection of the map. Bottom shows flood depths along the river when the depth from all stream cells are utilized. Notice that some areas shown as flooded in top figure are not flooded in bottom figure due to the influence of stream cells with lower depth calculations.**

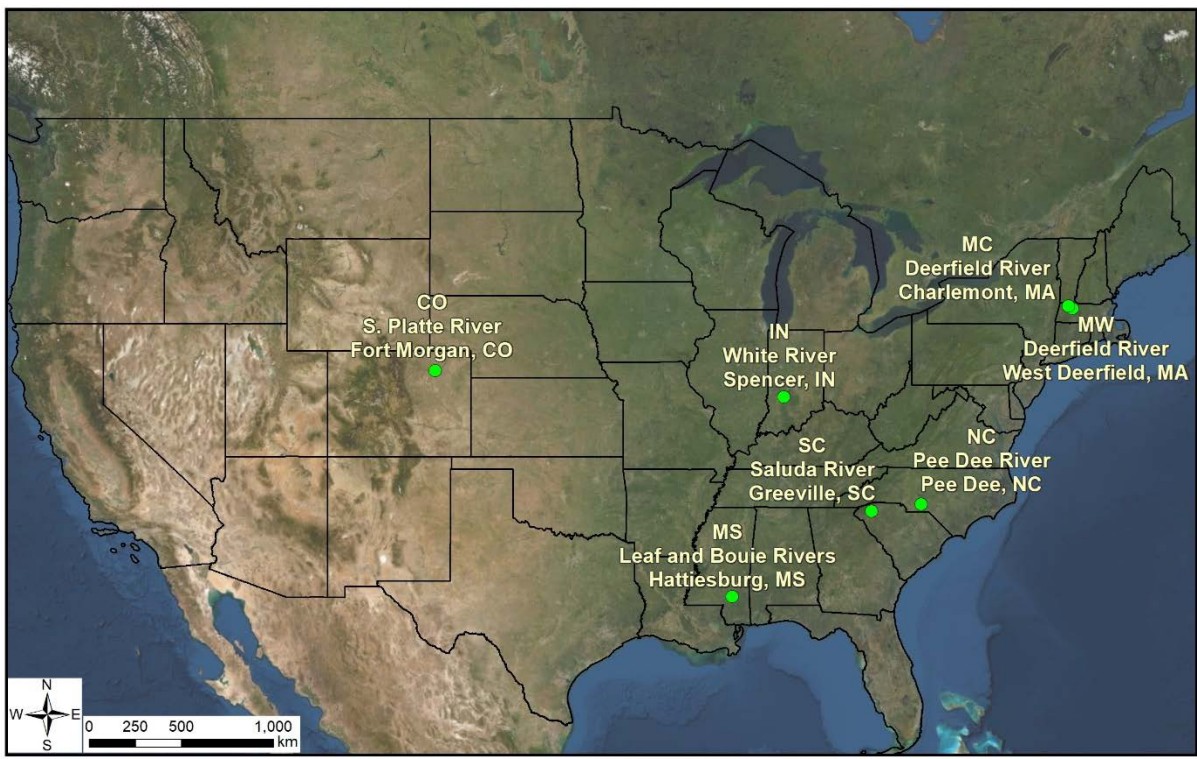

**Figure 3: USGS study sites used in this study. For each study site the location, ID, and river(s) are provided.**


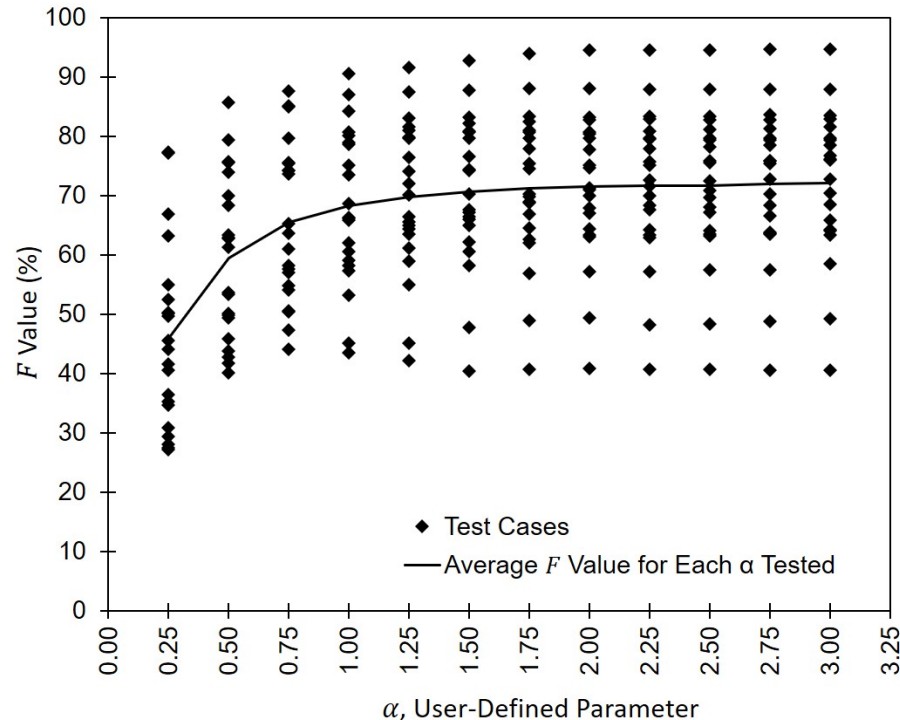

**Figure 4: Sensitivity analysis of the user-defined parameter $\alpha$, which controls the influence that each stream cell has on flooding the surrounding cells when using ARPP post-processing. F-statistic ($F$, percentage) values calculated using the observed and simulated flood inundation areas are plotted against the $\alpha$ value used in the simulation. In total, 252 simulations are shown (seven test sites, three flow scenarios, and twelve $\alpha$ values).**


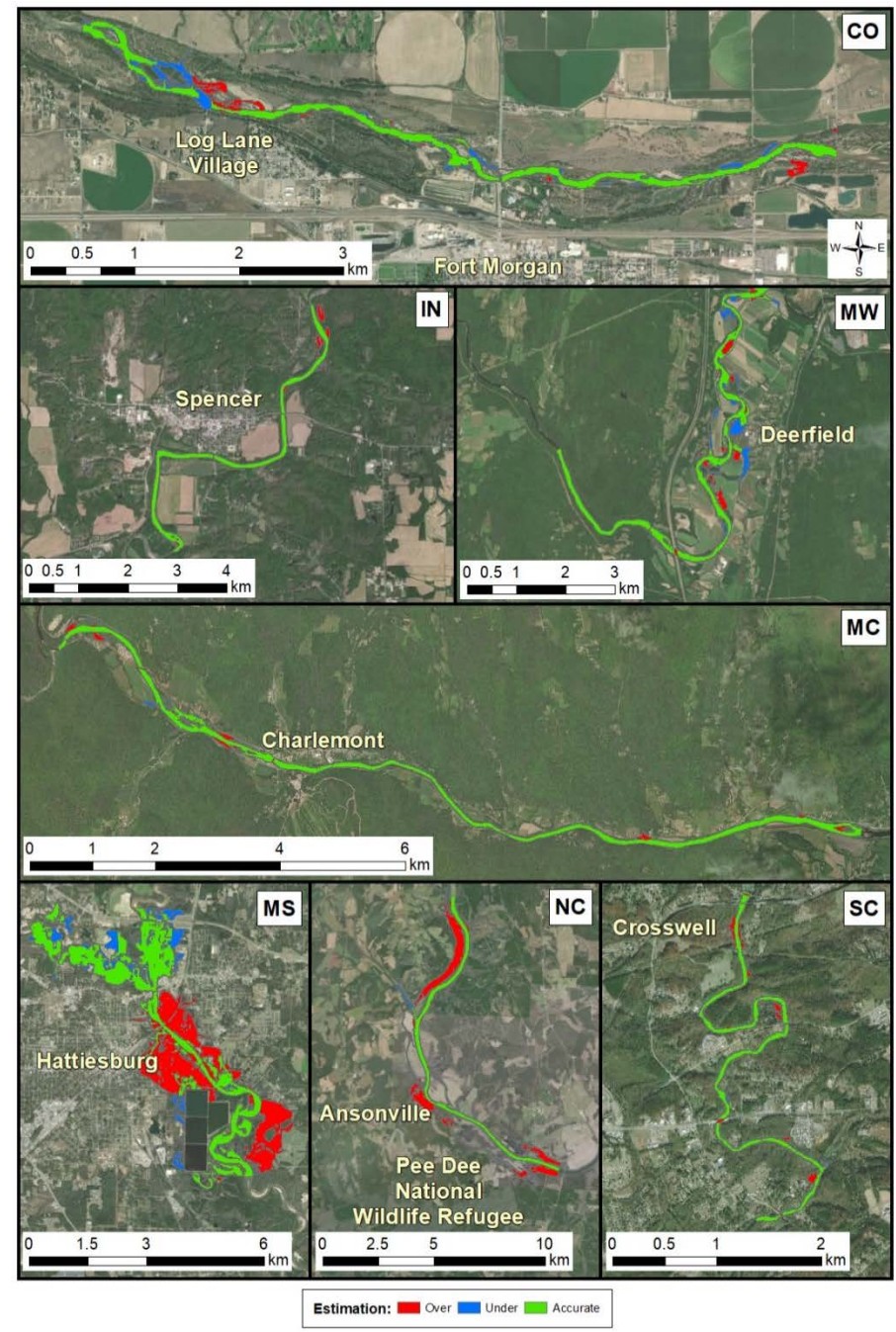

**Figure 5: Flood map comparison between AutoRoute+ARPP simulations and USGS flood maps for low flow events at the seven test sites. Areas shaded green (Accurate) indicate areas where AutoRoute+ARPP and the USGS flood maps agree. Areas shaded red (Over) indicate where only AutoRoute+ARPP simulates the area as flooded. Areas shaded blue (Under) indicate where only the USGS shows the area as flooded.**

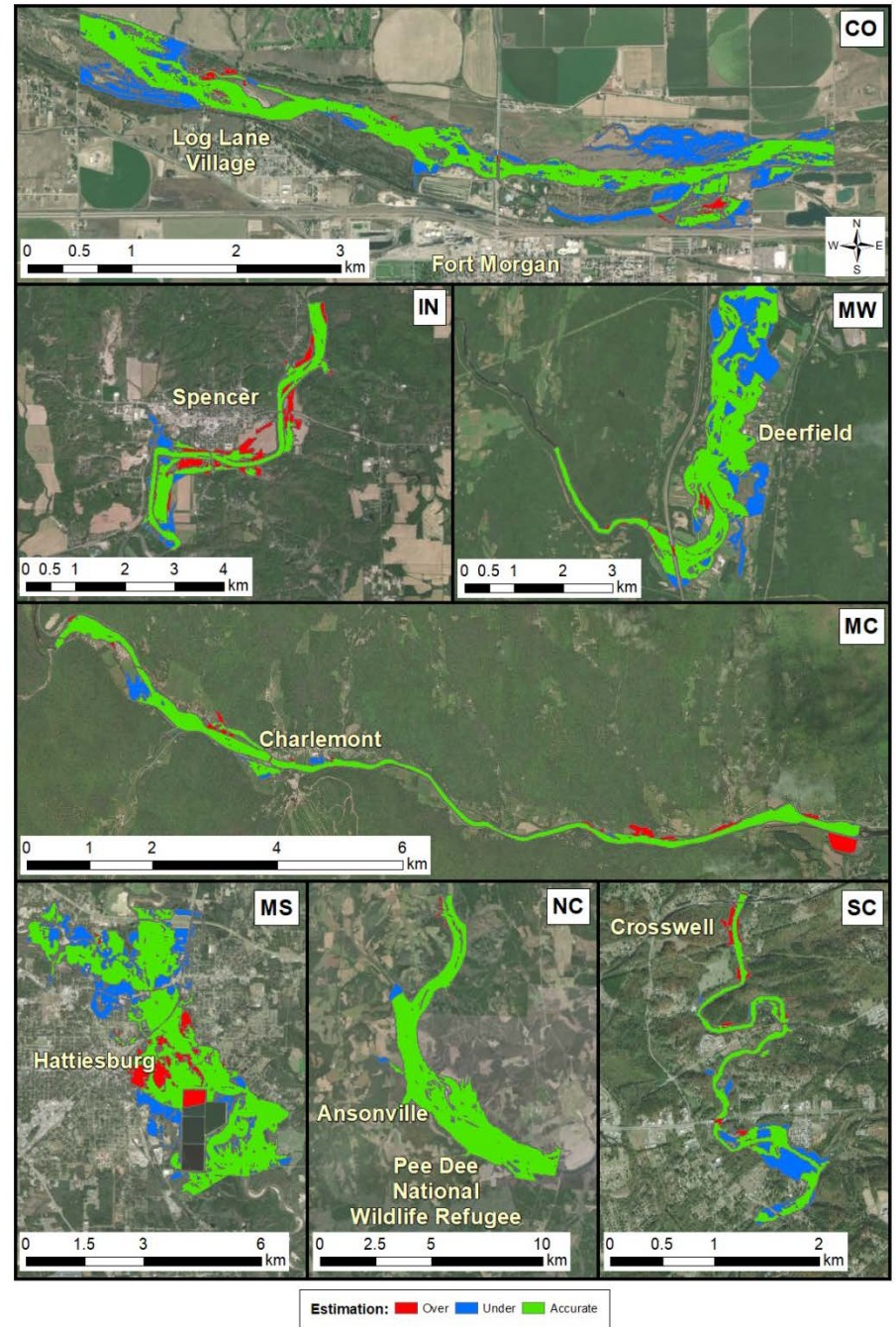


**Figure 6: Flood map comparison between AutoRoute+ARPP simulations and USGS flood maps for medium flow events at the seven test sites. Areas shaded green (Accurate) indicate areas where AutoRoute+ARPP and the USGS flood maps agree. Areas shaded red (Over) indicate where only AutoRoute+ARPP simulates the area as flooded. Areas shaded blue (Under) indicate where only the USGS shows the area as flooded.**


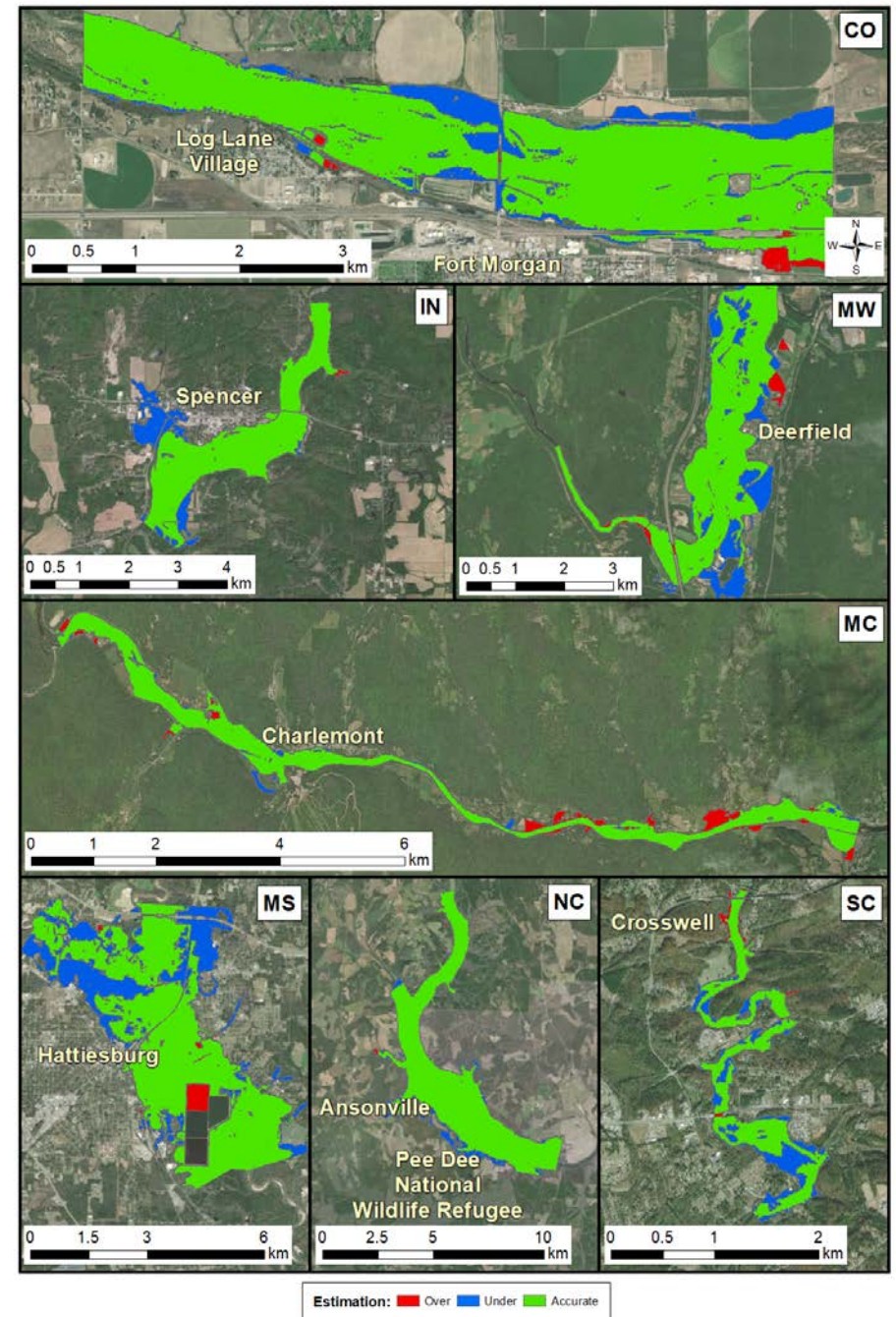

**Figure 7: Flood map comparison between AutoRoute+ARPP simulations and USGS flood maps for high flow events at the seven test sites. Areas shaded green (Accurate) indicate areas where AutoRoute+ARPP and the USGS flood maps agree. Areas shaded red (Over) indicate where only AutoRoute+ARPP simulates the area as flooded. Areas shaded blue (Under) indicate where only the USGS shows the area as flooded. Some of the overestimation in the MS model simulation occurs at water treatment ponds, which were not included in the USGS flood maps and can bias the results.**


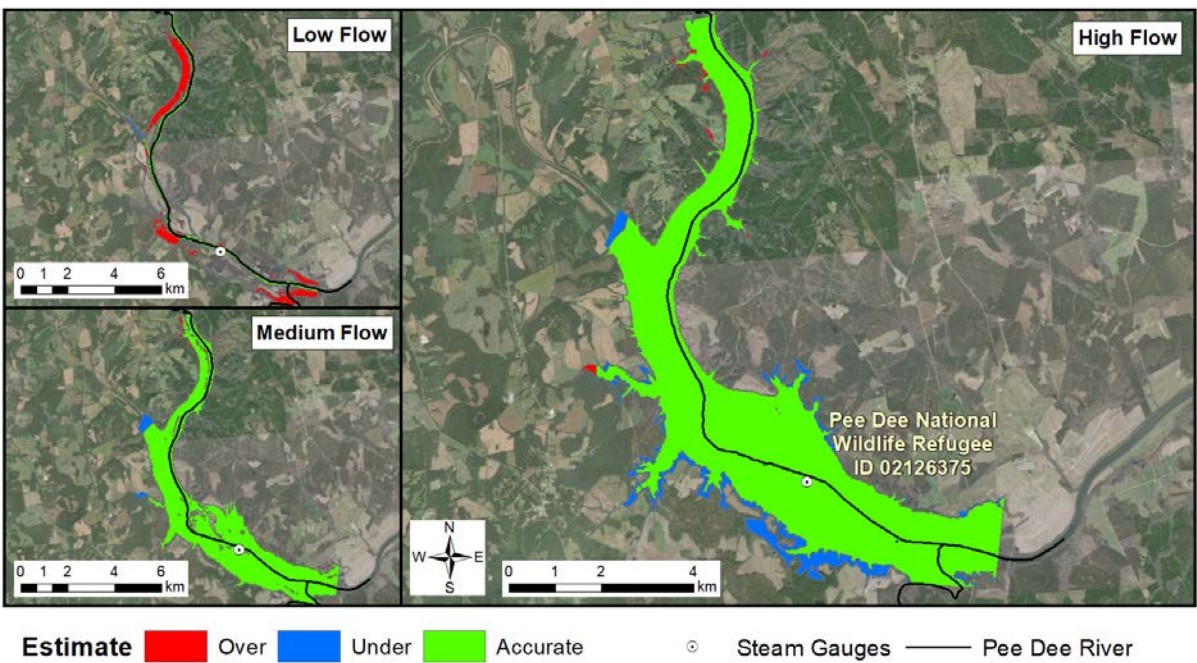

**Figure 8: Pee Dee, NC (NC-3m) flood map comparison between AutoRoute+ARPP simulations using ~3 m DEM and USGS flood maps. Areas shaded green (Accurate) indicate areas where AutoRoute+ARPP and the USGS flood maps agree. Areas shaded red (Over) indicate where only AutoRoute+ARPP simulates the area as flooded. Areas shaded blue (Under) indicate where only the USGS shows the area as flooded.**

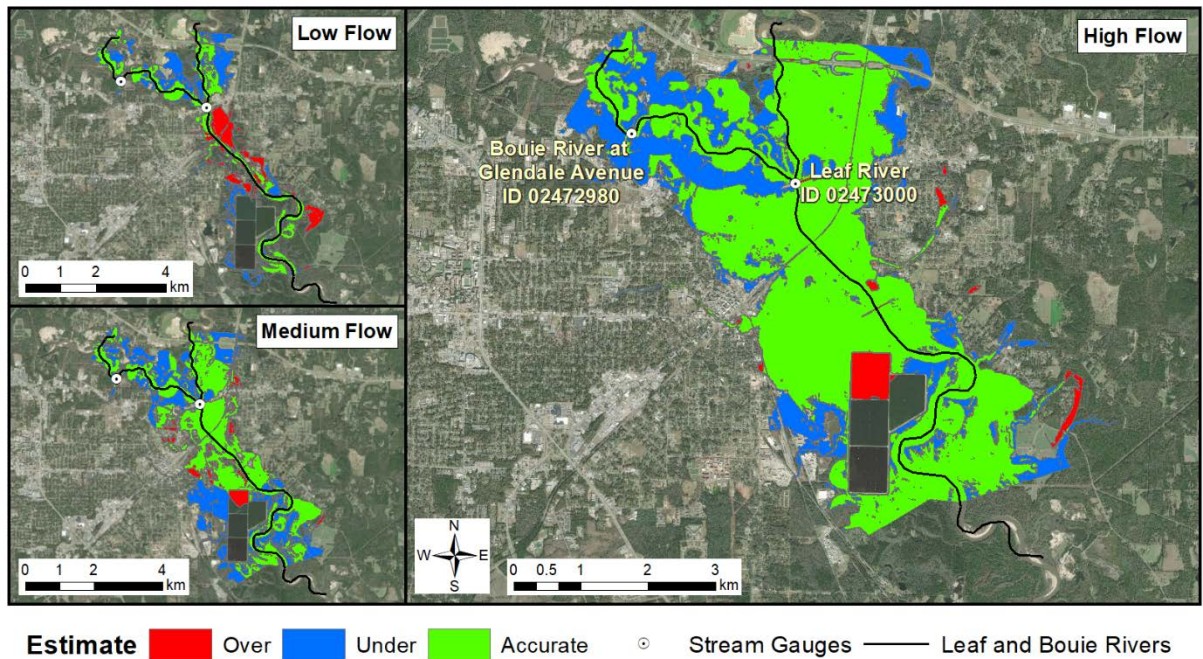


**Figure 9: Hattiesburg, MS (MS-3m) flood map comparison between AutoRoute+ARPP simulations using ~3 m DEM and USGS flood maps. Areas shaded green (Accurate) indicate areas where AutoRoute+ARPP and the USGS flood maps agree. Areas shaded red (Over) indicate where only AutoRoute+ARPP simulates the area as flooded. Areas shaded blue (Under) indicate where only the USGS shows the area as flooded. Some of the overestimation in the model simulation occurs at water treatment ponds, which were**
**not included in the USGS flood maps and can bias the results.**