# Peer review of "Improved Accuracy and Efficiency of Flood Inundation Mapping of Low-, Medium-, and High-Flow Events Using the AutoRoute Model"

_Natural Hazards and Earth System Sciences, 2019_

## Referee Comment (RC1) · Anonymous Referee #1 · 13 Jul 2019

The scope of this draft contains two parts, the first is to evaluate the performance of AutoRoute in different-flow-magnitude scenarios, which is the application part, and the second is to present the improvement of AutoRoute's computational efficiency brought by recent development on its post processing components, which is the development part. However, the current title of this paper only reflects the former one. The authors may consider rephrasing the title to make it more comprehensive. The application part of this draft across seven sites is solid but redundant figures are presented (see comment 8 for more details); the development part is inadequate to be reported as a significant progress. The innovation in method is not adequate to be presented as another paper adding upon the existing AutoRoute papers. Section 2.1 restates what has

been done previously, and only Section 2.2 on the AutoRoute post-processing script (ARPP) is the contribution of this draft. However, it only introduces a water surface elevation interpolator using IDW followed by a local depth computation step using the interpolated water surface and the raw terrain. I don't think the technique of ARPP is advanced enough to be marked as a new version of the tool. Overall, I think the content in the current draft is not enough to be reported as a separate scientific contribution that is different from previous AutoRoute publications. Two potential aspects the authors may consider to add to the current draft are (1) a stable solution for fixing the outliers during streamline water depth estimation process; (2) rerun the models on a few sites using the USGS lidar DEM instead of the NED and examine the corresponding change in performance metrics (F&E). Due to these concerns, a major revision decision is recommended to the editors. A set of technical issues and comments for the paper are provided here: 1. Line 15 The use of "accurate" (average F value of 63.3%) doesn't seem rigorous. Above what F score level, can the results be identified as "accurate"? Is that any reference for this definition? 2. Line 44-46 & 55-57 "However, this process relies heavily on pre-computed flow-depth relationships that may be difficult to apply in areas without highresolution DEM or NHD datasets." "Outside the U.S. stream networks (polyline format) for approximately 70% of the world have been created using HydroSHEDS and HydroBASINS datasets (Lehner and Grill, 2013)(see Snow et al., 2015 for an example)." Since both HAND and AutoRoute are raster-based low-complexity models with similar mechanism, the application stated in Line 55-57 overturn the statement in Line 44-46 about the limitation of the HAND approach. The HAND approach should also be able to applied when a regional hydrologic model is set up outside of the U.S. (https://tethys.servirglobal.net/apps/nepal-hiwatviewer/). 3. Figure 2(b) Within the channel zone, there are parts with shallower depth (compared to the rest), what's the reason for this variation? If the channel bottom elevation drops at a constant rate from upstream to downstream, when the water depth varies up and down as shown because they are computed separately cross section by cross section, how can AutoRoute ensure a smooth water surface profile along the streamline? 4.

Table 2 The medium flow for Site IN and Site CO reports the same number. Is this a coincidence, or a typo? 5. Line 173-183 Add a figure with different subplots showing the river network, site location and watershed of each site will be helpful to help readers understand the layout. 6. Line 41 & Line 186 The same NED is reported with different accuracy in different contexts (Line 41 ∼10m vs Line 186 ∼9m). Please change the numbers to a consistent value. 7. Line 186-187 "land cover classifications were obtained from the 2011 National Land Cover Database" Does the land cover raster share the same resolution as the elevation, or a resample process is introduced here? Please clarify. 8. Figure 3-Figure 9 I would rather treat them as 9 subplots of one figure, because they all express the same kind of information. Each plot of a figure in a journal article should have its unique point to present. The results in Table 3 and a figure with subplots showing results at three or four sites should be enough for the readers to get the points. Using the same figure template to make a separate plot for each site is ineffective and mindless. What will you do if you are testing over twenty sites, or over a region-scale river network? If there are a few sites with unique environment settings that result in a different-level accuracy, a separate figure should be created zooming into the spot of interest to demonstrate the point such as the impact of low-lying areas near the river or flat terrains. 9. Line 219-220 "Visually, NC (Figure 6) and MS (Figure 7) have the greatest amount of overestimation during the low flow event, resulting in the lowest ðİŘź values of all the simulations." Rather than only presenting the ratio-type performance metrics F & E, adding other columns for the actual inundation coverage area numbers will help readers get a better sense of the over/under estimation level.

---

## Referee Comment (RC2) · Anonymous Referee #2 · 8 Aug 2019

The manuscript by Follum et. al. titled, 'Flood Inundation Mapping of Low-, Medium-, and High-Flow Events Using the AutoRoute Model' presents a comparison of performance of the AutoRoute model for multiple flows ranging from low- to high flows. The study also presents an improvement in the existing model which enables smother flood extents without pits at a higher computational efficiency. However, the improvement in the performance of simplistic 1D models at higher flows (extreme events) and relative inaccuracy at lower flows is well known and quantified in several studies. As an example, please refer to the manuscript titled, 'Assessing the effect of different bathymetric models on hydraulic simulation of rivers in data sparse regions' by Dey et. al. 2019 which provides a similar comparison. I would recommend the authors to test

the model across these reaches using a finer resolution DEM (e.g. LIDAR) and evaluate the relative computational efficiency and accuracy using LIDAR vs NED. Also, in several places, the study suggests that river bathymetry is estimated using a simplistic exponential method which may not yield accurate results. I am wondering if the authors can compare the results from the AutoRoute model using a DEM that has bathymetry burned in it. Considering the authors used USGS flood maps for comparison, surveyed cross-sections are typically available for those sites through USGS. It would not only provide additional validation on the AutoRoute modeling for low flows but may also provide useful findings. That may help in adding research value to this manuscript. I am not sure if an improvement in the model simulation time from 35 minutes to 20.5 minutes is significant enough. Considering AutoRoute produces a single flood map for a specific flow, the authors should provide more discussion on how these results may scale up both in space and time. The improved computational efficiency may be more useful when simulating continuous flood maps at an hourly time scale for a regional-scale model or at a daily time scale for a continental-scale model. The authors should discuss this more in the manuscript. Although the application of the new post-processing procedure helps in improving the computational efficiency, the manuscript in its current form does not offer substantial analysis to warrant publication, and therefore, a major revision is needed. Specific comments Paragraph 95: Lines, "For high flow events the bathymetry in smaller streams can often be ignored...". Please provide references for this and the following statement. One example could be Dey et. al. 2019. Line 295: 'Although the flood inundation results for low-flow events are accurate (average ðÌŘź value of 63.3%)...'. I would be careful in calling an average F-value of 63.3% as accurate.

---

## Author Comment (AC1) · 16 Aug 2019

We would like to thank Referee #1 for their comments and suggestions. We hope this response will address their suggestions and lead to an improved manuscript.

General Issue #1, rephrasing of the title to capture both the development and application parts of the manuscript. We agree with the reviewer that the title needs to be rephrased to incorporate the "development" part of the manuscript. We plan to change the title to: "Improved Accuracy and Efficiency of Flood Inundation Mapping of Low-, Medium-, and High-Flow Events Using the AutoRoute Model".

[Figure]

General Issue #2, the use of a water surface elevation interpolator using IDW is the only new component to AutoRoute and not enough to warrant a separate publication. Although simple, the use of the water surface elevation interpolator produces improved accuracy (filling of holes in floodplain) and computational efficiency of the AutoRoute model. However, it is not the only new component described in this paper. The use of an automated bathymetric estimation within the AutoRoute is also new and is actually more important to the original application of AutoRoute as a connection between hydrologic data and mobility analysis (McKinley et al., 2012). In the revised paper we will emphasize the automatic bathymetric profile component within AutoRoute and it's application for mobility analysis.

General Issue #3, implement a stable solution for fixing the outliers during streamline water depth estimation process. We agree with the reviewer that a method to omit outliers will improve the accuracy of the flood inundation maps. We have recently developed a method within the AutoRoute post-processing script (ARPP) to omit individual outliers by analyzing the calculated depth along the entire reach of the river. In the revised manuscript we will implement and test this new method.

General Issue #4, rerun flood models using LiDar elevation instead of NED elevation. This suggestion was also made by Referee #2. In the revised paper we will test the use of LiDar at a few of the test sites and compare accuracy and computational efficiency.

Specific Issue #1, what constitutes an "accurate" F value? In the revised paper we will provide referenced criteria for what constitutes an accurate F value.

Specific Issue #2, differences between HAND and AutoRoute. The referee is correct that the current manuscript does not highlight the differences between the HAND and AutoRoute model. A more robust description of the similarities and differences between the two models will be added to the revised paper that highlight the need for AutoRoute models when connecting to mobility models (McKinley et al., 2012).

Specific Issue #3, variations in flow depth along river reach in Figure 2b. The referee

is correct that the variations in depth in Figure 2b are caused by individual depth calculations at each cross-section. The implementation of a stable solution for omitting outliers (General Issue #3) in the revised paper will likely remove the large variations in depth along the river reach.

Specific Issue #4, same medium flow rates at IN and CO sites in Table 2. The flow rates for IN (Nystrom, 2013) and CO (Kohn and Patton, 2018) shown in Table 2 are correct.

Specific Issue #5, addition of a figure showing the river networks for the NC and MS sites. A figure showing the streamlines and gage locations for the NC and MS sites will be included in the revised paper.

Specific Issue #6, different spatial resolution reported for NED data. Line 41 will be changed in the revised paper to show the spatial resolution of NED data to be ∼9m to match Line 186.

Specific Issue #7, does the land cover dataset have the same resolution as the elevation dataset? The land cover dataset has a spatial resolution of approximately 30m, and is therefore resampled to the same resolution as the elevation dataset. This will be better described in the revised paper.

Specific Issue #8, redundant figures. In the revised paper we plan to consolidate the flood inundation results (Figures 3-9). The goal of the consolidation will be to omit unnecessary figures/subplots, thus focusing on the flood maps highlighted in the results and discussion.

Specific Issue #9, include inundation coverage areas in Table 3. In the revised paper we will include coverage areas (accurately-simulated area, under-simulated area, and over-simulated area) in both Table 3 as well as the results and discussion sections.

Kohn, M. S. and Patton, T. T.: Flood-inundation maps for the South Platte River at Fort Morgan, Colorado, U.S. Geological Survey Scientific Investigations Report 2018–5114,

Available from: https://doi.org/10.3133/sir20185114, 2018.

McKinley, G. B., Mason, G. L., Follum, M. L., Jourdan, M. R., LaHatte, C. W. and Ellis, J.: A Route Corridor Flood Vulnerability System, Geotechnical and Structures Laboratory Technical Report ERDC/GSL TR-12-29. U.S. Army Engineer Research and Development Center, Geotechnical and Structures Laboratory, Vicksburg, Mississippi, 2012.

Nystrom, E. A.: Flood-inundation maps for the White River at Spencer, Indiana, U.S. Geological Survey Scientific Investigations Map 3251, Available from: http://pubs.usgs/gov/sim/3251/, 2013.

―――――――――――――――――――

---

## Author Comment (AC2) · 16 Aug 2019

We would like to thank Referee #2 for their comments and suggestions. We hope this response will address their suggestions and lead to an improved manuscript.

General Issue #1, testing of low and high flow events within 1-D hydraulic models has previously been completed. This paper aims to improve the accuracy and computational efficiency of the AutoRoute model in simulating low-, medium-, and high-flow events. AutoRoute is somewhat unique as it has been applied at the regional- to continental-scale by the U.S. military, but has only been tested against high-flow events (Lines 50-54 and 79-82). We agree with the referee that 1-D hydraulic models often

perform better during high flow events, which this study found as well (Line 296). In the revised paper we will add references to this point, including Dey et al. (2019).

General Issue #2, provide more discussion on how AutoRoute flood simulations may scale up both in space and time. In the revised paper we will provide more discussion on how the improvements to the AutoRoute model will lead to more efficient flood mapping capabilities at the regional- and continental- scales. More information will also be provided on how the U.S. Army Coastal and Hydraulics Laboratory utilizes the AutoRoute model for both flood inundation and mobility assessments.

General Issue #3, improved computational efficiency may not warrant publication. Although simple, the use of the water surface elevation interpolator discussed in this paper produces improved accuracy (filling of holes in floodplain) and computational efficiency of the AutoRoute model. However, it is not the only new component described in this paper. The use of an automated bathymetric estimation within the AutoRoute is also new and is actually more important to the original application of AutoRoute as a connection between hydrologic data and mobility analysis (McKinley et al., 2012). In the revised paper we will emphasize the automatic bathymetric profile component within AutoRoute and its application for mobility analysis. We will also discuss how different bathymetric profile methods could be implemented, with a reference to Dey et al. (2019).

Specific Suggestion #1, rerun flood models using LiDar elevation with bathymetry estimations (potentially) already "burned in". This suggestion was also made by Referee #1. In the revised paper we will test the use of LiDar at a few of the test sites and compare accuracy and computational efficiency. We agree with the referee that comparing AutoRoute results using NED and LiDar may provide useful findings.

Specific Issue #1, provide reference for the line "For high flow events the bathymetry in smaller streams can often be ignored...". In the revised paper we will remove this sentence and the following sentence that states that bathymetry is more important in

low-flow simulations (Lines 98-100). These sentences will be replaced with a sentence emphasizing the importance of bathymetry in flood model simulations. This new sentence will have references, including Dey et al. (2019).

Specific Issue #2, what constitutes an "accurate" F value? In the revised paper we will provide referenced criteria for what constitutes an accurate F value.

Dey, S., Saksena, S., Merwade, V.: Assessing the Effect of Different Bathymetric Models on Hydraulic Simulation of Rivers in Data Sparse Regions, Journal of Hydrology, 575, 838-851, 2019.

McKinley, G. B., Mason, G. L., Follum, M. L., Jourdan, M. R., LaHatte, C. W. and Ellis, J.: A Route Corridor Flood Vulnerability System, Geotechnical and Structures Laboratory Technical Report ERDC/GSL TR-12-29. U.S. Army Engineer Research and Development Center, Geotechnical and Structures Laboratory, Vicksburg, Mississippi, 2012.
* * *

---

## Author Response (AR1)

**Improved Accuracy and Efficiency of Flood Inundation Mapping of Low-, Medium-, and High-Flow Events Using the AutoRoute Model**

Michael L. Follum[1,2], Ricardo Vera[3], Ahmad A. Tavakoly[1,4], and Joseph L. Gutenson[1,5]

[1]Coastal and Hydraulics Laboratory, Engineer Research and Development Center, 3909 Halls Ferry Road, Vicksburg, MS 39180, USA
[2]Wyoming Area Office, U.S. Bureau of Reclamation, 705 Pendell Blvd., Mills, WY 82644, USA.
[3]Cold Regions Research and Engineering Laboratory, Engineer Research and Development Center, 72 Lyme Road, Hanover, NH 03755, USA
[4]Earth System Science Interdisciplinary Center, University of Maryland, College Park, MD 20740, USA
[5]National Water Center, National Oceanic and Atmospheric Administration, 205 Hackberry Ln, Tuscaloosa, AL 35401, USA

*Correspondence to*: Michael L. Follum (follumm@gmail.com)

**Summary of Author Response**

We would like to thank both referees (R1 and R2) for their comments and suggestions. We appreciate all their insights, which we believe improved the manuscript. In this document we respond to each comment and describe the associated changes to the manuscript. All line and page numbers refer to the manuscript with no changes marked. The point-by-point response to the reviews are presented first, followed by a blank page, and lastly followed by the marked-up version of the manuscript.

R1 - General Issue #1, rephrasing of the title to capture both the development and application parts of the manuscript. We agree with the reviewer that the title needed to be rephrased to incorporate the "development" part of the manuscript. The title has now been changed to:

"Improved Accuracy and Efficiency of Flood Inundation Mapping of Low-, Medium-, and High-Flow Events Using the AutoRoute Model".

R1 & R2 - General Issue #2, the use of a water surface elevation interpolator using IDW is the only new component to AutoRoute and not enough to warrant a separate publication. Although simple, the use of the water surface elevation interpolator produces improved accuracy (filling of holes in floodplain while accounting for terrain) and computational efficiency of the AutoRoute model. However, it is not the only new component described in this paper. The use of an automated bathymetric estimation within the AutoRoute is also new (Lines 52-53, 87-89, 102-110, Figure 1) and may actually be more important to the original application of AutoRoute as a connection between hydrologic data and mobility analysis (Lines 36-41, 51-52). The revised paper now provides more emphasis on the automatic bathymetric profile component within AutoRoute and its application for mobility analysis (Lines 318-319, 349-351). A future research area now highlighted is the

use of different bathymetric profile methods to improve flood and mobility analysis (includes reference to Dey et al., 2019)

35      (Lines 349-351).

R1 - General Issue #3, implement a stable solution for fixing the outliers during streamline water depth estimation process.
We agree with the reviewer that a method to omit stream depth outliers will improve the accuracy of the flood inundation
maps. We tested two simple methods to remove outliers based on average depth of a stream reach and average water surface

40      elevation of a stream reach. Stream cell calculations were omitted if outside of 1 or 2 standard deviations from the average
depth or surface elevation of the stream reach. Neither method improved the flood inundation results and often increased the
error in certain locations along the stream reach. Lines 157-161 have been modified to better describe the affect that outliers
have on both flood inundation and mobility assessments. Removal of outliers in the stream water depth estimation process is
now included as an avenue of future research (Lines 160-161, 351-352).

45

R1 & R2 - General Issue #4, rerun flood models using LiDar elevation instead of NED elevation. We agree with the reviewers
that a test using higher-resolution elevation data would be beneficial. We added a new section (Section 4.2) to use 3-m
elevation data and reran the model for the MS and NC test cases. Results are explained in Lines 261-278, 290-291, 335-340,
and Figures 6 and 7.

50

R2 - General Issue #1, testing of low and high flow events within 1-D hydraulic models has previously been completed. This
paper aims to improve the accuracy and computational efficiency of the AutoRoute model in simulating low-, medium-, and
high-flow events. AutoRoute is somewhat unique from other flood-inundation models as it has been applied at the regional-
to continental-scale by the U.S. military for both flood inundation and mobility purposes (Lines 36-41, 54-58), but has only

55      been tested against high-flow events (Lines 15, 82-85). With the noted exception of Wing et al. (2017) (Lines 28-30), most
studies that analyze low- and high-flow events are doing so at the reach-scale using one-dimensional (1-D) hydraulic models
developed for reach- to small basin-scale applications (e.g. Dey et al., 2019; Tayefi et al., 2017). Therefore, analyzing low-,
medium-, and high-flow events within a model developed to operate using high-resolution elevation data at the regional- to
continental-scale is new.

60      We agree with the referee that 1-D hydraulic models often perform better during high flow events, which this study found as
well (Line 245-247) and we now reference a new publication (Dey et al. 2019).

R2 - General Issue #2, provide more discussion on how AutoRoute flood simulations may scale up both in space and time.
We agree with the reviewer that more context was needed for how AutoRoute could be implemented at larger scales. The text

65      was modified to describe how the U.S. Army Coastal and Hydraulics Lab currently uses the Streamflow Prediction Tool (SPT)
and AutoRoute to provide hydrologic and mobility awareness for approximately 70% of the world (Lines 36-41). Lines 306-
309 now discuss how the increased efficiency of the AutoRoute model will enable better use of the tool for applications outside

of the United States. Lines 309-316 and 352-354 describe how future adaptions to AutoRoute could further improve the computational efficiency of the model, enabling it to be used for larger areas.

70

R1 & R2 - Specific Issue #1, what constitutes an "accurate" F value? To the best of the authors' knowledge, the threshold of an accurate F value has not been defined in literature. Usually, "F" values are compared relative to scenarios defined in each project (e.g. Model A results have a higher F value than Model B results). Terrain and geographic complexity of the region also affect what is considered an accurate F value. For example, in regions with high topography where channels are well-

75 defined one would expect a one-dimensional model to produce a higher F value than for a shallow river in a coastal plain. Because of this ambiguity, we have modified the text in Lines 16, 20, 225-227, and 329 to reflect that the F values produced in this study are comparable with published values in literature.

R1 - Specific Issue #2, inconsistency in limitations of HAND. The referee is correct that the limitations of the HAND method

80 to operate outside of the United States were incorrectly characterized. We revised Lines 45-49 to describe the limitation of HAND needing flow-depth relationships that are currently not available for much of the world. Lines 310-313 and 352-354 describe how AutoRoute could potentially be used to create these flow-depth relationships that in-turn could be used with the HAND method.

85 R1 - Specific Issue #3, variations in flow depth along river reach in Figure 2b. The referee is correct that the variations in depth in Figure 2b are caused by individual depth calculations at each cross-section. As discussed in a previous comment, a solution for omitting outliers was tested but was not successful and is therefore left for consideration in future work (Lines 160-161, 351-352). Therefore, we revised/added the text (Lines 157-164) to discuss the variations in depth and how they mainly affect the depth values in the center of the channel. The text also addresses how the variations in depth will likely have

90 a greater affect on mobility analysis when using AutoRoute.

R1 - Specific Issue #4, same medium flow rates at IN and CO sites in Table 2. The flow rates for IN (Nystrom, 2013) and CO (Kohn and Patton, 2018) shown in Table 2 have been checked and are correct.

95 R1 - Specific Issue #5, addition of a figure showing the river networks for the NC and MS sites. Figures 6 and 7 (referenced in Line 185 & 189) now show the streamlines and gage locations for the NC and MS sites, respectively.

R1 - Specific Issue #6, different spatial resolution reported for NED data. All references to a "10 m" spatial resolution for the 1/3 arc-second National Elevation Dataset have been changed to "9 m".

100

R1 - Specific Issue #7, does the land cover dataset have the same resolution as the elevation dataset? The land cover dataset has a spatial resolution of approximately 30 m (Line 195-196) and is therefore resampled to the same resolution as the elevation dataset (Lines 195-196, 265-266).

105 R1 - Specific Issue #8, redundant figures. We agree with the reviewer's suggestion and Figures 3-9 in the original manuscript have now been consolidated into Figures 3-5 (reduction of 4 figures).

R1 - Specific Issue #9, include inundation coverage areas in Table 3. We revised Table 3 to include coverage areas (observed area, simulated area, accurately-simulated area, under-simulated area, and over-simulated area). References to the coverage

110 areas are now made in the Results and Discussion section.

R2 - Specific Issue #1, provide reference for the line "For high flow events the bathymetry in smaller streams can often be ignored...". We revised the text (Lines 102-106) to emphasize the importance of bathymetry in flood model simulations and mobility assessments.

115

**References**

[revised manuscript text omitted]

---

## Author Response (AR2)

**Improved Accuracy and Efficiency of Flood Inundation Mapping of Low-, Medium-, and High-Flow Events Using the AutoRoute Model**

Michael L. Follum[1,2], Ricardo Vera[3], Ahmad A. Tavakoly[1,4], and Joseph L. Gutenson[1,5]

[1]Coastal and Hydraulics Laboratory, Engineer Research and Development Center, 3909 Halls Ferry Road, Vicksburg, MS 39180, USA
[2]Wyoming Area Office, U.S. Bureau of Reclamation, 705 Pendell Blvd., Mills, WY 82644, USA.
[3]Cold Regions Research and Engineering Laboratory, Engineer Research and Development Center, 72 Lyme Road, Hanover, NH 03755, USA
[4]Earth System Science Interdisciplinary Center, University of Maryland, College Park, MD 20740, USA
[5]National Water Center, National Oceanic and Atmospheric Administration, 205 Hackberry Ln, Tuscaloosa, AL 35401, USA

*Correspondence to*: Michael L. Follum (followm@gmail.com)

**Summary of Author Response**

We would like to thank the editor (ED) and referees (R1, R2, and R3) for their comments and suggestions.  We appreciate all their insights, which we believe improved the manuscript.  In this document we respond to each comment and describe the associated changes to the manuscript.  All line and page numbers refer to the manuscript with no changes marked.

ED – General Issue #1.  Accuracy comparison between the old post-processing procedure and the updated post-processing scripts.

We agree with the editor that including a comparison between the previous post-processing procedure (GISPP) and the updated post-processing procedure (ARPP) will improve the paper.  We have modified the beginning of Section 4.1 (Lines 236-249) to incorporate a test between the GISPP and ARPP.  The results ($F$ and $E$ statistics) are now shown in Table 3.  In several locations we now also discuss the improvement in accuracy using the ARPP (Lines 16, 359-366).

ED – General Issue #2.  Parameter sensitivity analyses on post-processing parameters.

Lines 224-233 and Figure 4 have now been added to provide a parameter sensitivity test on the user-defined $\alpha$ value, which in ARPP controls the influence that each stream cell has on flooding the surrounding cells.

R1 - General Issue #1. More details should be given on how the parameters of the exponential channel profile is generated from a given top width.

We agree with the reviewer and a section has been added that describes the method and parameters required for AutoRoute to estimate a bathymetric profile (Lines 111-121). Figure 1 has also been modified to better explain the methods for bathymetric estimation within AutoRoute.

R1 - General Issue #2. The authors should at least list a few possible reasons why these outliers occur (question associated with Figure 2).

Outliers in the flood depth calculation exist for several reasons, many of them are listed in Lines 69-73. Lines 169-170 were also added to address the source of outliers. We agree with the reviewer that the existence of depth outliers needs to be resolved and this topic is listed as an avenue for future research (Lines 387-389).

R1 - General Issue #3. Does the lidar DEM share the same flat bottom width as the 9-meter NED? Also, another figure showing the difference between the ned-simulated inundation extent and the lidar-simulated extent might be desirable. Two subplots of the raw DEMs for the same spot can be added on the left as the reference to give readers an idea about the level of land surface details different-resolution DEMs contains.

We agree with the reviewer that analysing the effects of different elevation resolution on flood inundation accuracy is important and we now include Lines 291-295 to stress the importance of quality elevation data. Several papers have addressed the issue of elevation resolution and accuracy when using hydraulic models (a few are now listed in Lines 293-294). However, none of these research efforts have included the AutoRoute model and therefore we have included analysis of elevation and bathymetry data as a future research effort (Lines and 385-386).

In Section 4.2 and Figures 8 and 9 we utilize 3-m elevation data, which is processed from LiDar. We have used raw LiDar in previous studies and have found the spikes in the data make it difficult to use within the AutoRoute model (numerous outliers related to R1 - General Issue #2). To specifically answer the reviewer's question, both the 9- and 3-m elevation datasets used in this study have "flat" bottoms over the water surface. Depending on the (post)processing of other LiDar datasets, the water surface may or may not be flat. In order to keep the scope of the paper limited and to minimize the number and size of the Figures, we have included several references to studies that have analysed various elevation datasets (Lines 293-294) in-lieu of adding/modifying Figures. We reiterate that we do see the value of using various elevation datasets but believe that this would be better served in a future research effort.

R2 – General Issue #1. A figure providing the geographic locations of all the sites used in the study on a US map to provide more context to the readers outside the United States.

We agree with the reviewer and now included Figure 3 (also see Line 184), which shows the test site locations on a US map.

R3 – General Issue #1. Accuracy comparison between the old post-processing procedure and the updated post-processing scripts.

We agree with the reviewer that including a comparison between the previous post-processing procedure (GISPP) and the updated post-processing procedure (ARPP) will improve the paper. We have modified the beginning of Section 4.1 (Lines 236-249) to incorporate a test between the GISPP and ARPP. The results ($F$ and $E$ statistics) are now shown in Table 3. In several locations we now also discuss the improvement in accuracy using the ARPP (Lines 16, 359-366).

R3 – General Issue #2. Why not presenting a tool that incorporates the whole AutoRoute post-processing workflow, but only a script is specifically created for weight calculation?

The only post-processing component not included within ARPP is the conversion from raster format to flood inundation polygons (for which one can use ArcGIS or GDAL) (Line 163). The reason we have not included the conversion from raster to polygon within ARPP is that this post-processing step is often not utilized (mobility models typically utilize raster-based inputs). In the next update to ARPP we will likely provide the option to convert directly to shapefile polygons using ARPP.

R3 – General Issue #3. Parameter sensitivity analyses on post-processing parameters.

Lines 224-233 and Figure 4 have now been added to provide a parameter sensitivity test on the user-defined $\alpha$ value, which in ARPP controls the influence that each stream cell has on flooding the surrounding cells.

R3 – General Issue #4. Utilize higher-elevation data.

We agree with the reviewer that analysing the effects of different elevation resolution on flood inundation accuracy is important and we now include Lines 291-295 to stress the importance of quality elevation data. Several papers have addressed the issue of elevation resolution and accuracy when using hydraulic models (a few are now listed in Lines 293-294). However, none of these research efforts have included the AutoRoute model and therefore we have included analysis of elevation and bathymetry data as a future research effort (Lines and 385-386).

In Section 4.2 and Figures 8 and 9 we utilize 3-m elevation data, which is processed from LiDar. We have used raw LiDar in previous studies and have found the spikes in the data make it difficult to use within the AutoRoute model (numerous flood depth outliers at the stream cells), however these errors associated with raw LiDar would not be unique to AutoRoute. In order to keep the scope of the paper limited and to minimize the number and size of the Figures, we have included several references to studies that have analysed various elevation datasets (Lines 293-294) in-lieu of simulating more results using even higher resolution elevation data.. We reiterate that we do see the value of using various elevation datasets but believe that this would be better served in a future research effort.

100 R3 – General Issue #5. was the computational cost reduction (described in Section 4.3) purely a result of the use of GDAL I/O functions (L477)?

The use of GDAL I/O did reduce the computation cost of running AutoRoute (Lines 332-335). However, the reduction of computation cost of the post-processing (the majority of total computational savings, Lines 332-338) is due to the use of the ARPP method outlined in Section 2.2 instead of using the GISPP method that relies on ArcGIS (ESRI, 2011) Boundary Clean
105 and Aggregate Polygons functions (Lines 143-148).

R3 – Minor Issue #1. L186 should be millions instead of thousands.

The reviewer is correct that this value could be millions but in some applications it is still thousands and so no edits were made.
110

R3 – Minor Issue #2. Citations for Streamflow Prediction Tool.

The Streamflow Prediction Tool (SPT) is used by the U.S. Army for streamflow predictions outside the Continental United States, while the National Water Model (NWM) is only for locations within the Continental United States. AutoRoute currently utilizes flow rates from the SPT but could also utilize flows from the NWM. Therefore, the citations to the SPT are
115 relevant and no edits were made.

R3 – Minor Issue #3. Unclear as to what does "AutoRoute is operated in an ad-hoc basis" mean. Authors need to clarify this statement. Also, why only 70% of the world has SPT implementations?

Lines 40-44 were editd/included to better define the role of AutoRoute in an ad-hoc basis as well as state why SPT is operational
120 for only ~70% of the world.

R3 – Minor Issue #4. A figure showing the geographic locations of these seven sites would be helpful to readers.

We agree with the reviewer and now included Figure 3 (also see Line 184), which shows the test site locations on a US map.

125 R3 - Minor Issue #4. Currently the authors only plainly presented statistics, but I suggest the authors discuss more about their difference. Is elevation the only difference that explain their different model performance?

Statistics were added to Section 4.1 and complexities due to terrain, land cover, topography, flow rates utilized, etc. are discussed as potential reasons for similarities and differences in model performance.

**References**

[revised manuscript text omitted]

---

## Author Response (AR3)

**Improved Accuracy and Efficiency of Flood Inundation Mapping of Low-, Medium-, and High-Flow Events Using the AutoRoute Model**

Michael L. Follum[1,2], Ricardo Vera[3], Ahmad A. Tavakoly[1,4], and Joseph L. Gutenson[1,5]

[1]Coastal and Hydraulics Laboratory, Engineer Research and Development Center, 3909 Halls Ferry Road, Vicksburg, MS 39180, USA
[2]Wyoming Area Office, U.S. Bureau of Reclamation, 705 Pendell Blvd., Mills, WY 82644, USA.
[3]Cold Regions Research and Engineering Laboratory, Engineer Research and Development Center, 72 Lyme Road, Hanover, NH 03755, USA
[4]Earth System Science Interdisciplinary Center, University of Maryland, College Park, MD 20740, USA
[5]National Water Center, National Oceanic and Atmospheric Administration, 205 Hackberry Ln, Tuscaloosa, AL 35401, USA

*Correspondence to*: Michael L. Follum (follumm@gmail.com)

**Summary of Author Response**

We would like to thank the editor for their comments and detailed suggestions. We have made all the edits requested (list shown below). All line and page numbers refer to the manuscript with no changes marked.

- Line 20, using twice the word 'improved' in one sentence. Maybe replace the 2nd 'improved' with 'enhanced'.
- Line 189, starting sentences on the same line with the same words 'Each study'. Consider rewrite to accommodate for this.
- Line 314, it is probably easier for the reader to write out 'constant ni' to 'Manning Roughness Coefficient (ni)', so that the reader is reminded what ni stands for.
- Line 385 – 394 (last paragraph of the conclusions). Consider rewriting so the word 'improve(d)(ment)' is less often used.

Additionally, we have added the required following sections:

- Code / Data Availability (Lines 402-405)
- Author Contribution (Lines 406-407)
- Competing Interests (Lines 408-409)
- Special Issue Statement (Lines 410-411)

[revised manuscript text omitted]